



# Linking rain into ice microphysics across the melting layer in stratiform rain: a closure study

Kamil Mróz[1], Alessandro Battaglia[2,3], Stefan Kneifel[4], Leonie von Terzi[4], Markus Karrer[4], and Davide Ori[4]

[1]National Centre for Earth Observation, University of Leicester, Leicester, UK
[2]University of Leicester, Leicester, UK
[3]Politecnico of Turin, Turin, Italy
[4]Institute for Geophysics and Meteorology, University of Cologne, Cologne, Germany

**Correspondence:** Kamil Mróz (kamil.mroz@le.ac.uk)

**Abstract.** This study investigates the link between rain and ice microphysics across the melting layer in stratiform rain systems using measurements from vertically pointing multi-frequency Doppler radars. A novel methodology to examine the variability of the precipitation rate and the mass-weighted melted diameter ($D_m$) across the melting region is proposed and applied to a 6h-long case study, observed during the TRIPEx-pol field campaign at the Jülich Observatory for Cloud Evolution Core Facility

and covering a gamut of ice microphysical processes. The methodology is based on an optimal estimation (OE) retrieval of particle size distributions (PSD) and dynamics (turbulence and vertical motions) from observed multi-frequency radar Doppler spectra applied both above and below the melting layer. The retrieval is first applied in the rain region; based on a one-to-one conversion of raindrops into snowflakes, the retrieved Drop Size Distributions (DSD) are propagated upward to provide a first guess for the snow PSDs. These ice PSDs are then used to constrain the OE snow retrieval where Doppler spectra are

simulated based on different snow models, which consistently compute fall-speeds and electromagnetic properties. The results corresponding to the best matching models are then used to compute snow fluxes and $D_m$, which can be directly compared to the corresponding rain quantities.

For the case study, the total accumulation of rain (2.65 mm) and the melted equivalent accumulation of snow (2.60 mm) show only a 2% difference. The analysis suggests that the mass flux through the melting zone is well preserved except the

periods of intense aggregation and intense riming where the precipitation rates were respectively larger and lower in ice than in the rain below. Moreover, it is shown that, the mean mass weighted diameter of ice is strongly related to the characteristic size of the underlying rain. With a simple scaling, $D_m^{ice} = 1.21 D_m^{rain}$, the characteristic size of snow can be predicted with a root-mean-square-error of 0.12 mm. This formula leads to slight underestimation of the ice size during aggregation, potentially due to the breakup of melting snowflakes, and to overestimation during riming where the additional particle growth within the

melting layer cannot be unambiguously attributed to one process. The proposed methodology can be applied to long-term observations to advance our knowledge of the processes occurring across the melting region; this can then be used to improve assumptions underpinning space-borne radar precipitation retrievals.



*Copyright statement.* TEXT

# 1 Introduction

The accurate quantification of ice cloud macro-physical (height, thickness) and micro-physical properties (characteristic particle size and shape, mass content and number concentration) is paramount for understanding the current state of Earth's hydrological cycle and energy budget and to improve the representation of clouds for climate model predictions (Stephens, 2005; Tao et al., 2010). Macro-physical properties can be well captured by active remote sensing instruments (Stephens et al., 2018); on the other hand, the characterization of ice microphysics remains one of the most challenging problems (Heymsfield

et al., 2018) because of the substantial number of assumptions about the PSD and the particle "habit" type (such as dendrites, columns, rosettes, aggregates, or rimed particles) required in remote sensing techniques. While the characterization of small ice crystals is particularly relevant for detailing the radiative effects of high ice clouds, understanding processes like aggregation, riming and deposition are essential for accurately modelling precipitation.

The study of stratiform precipitation encompasses the investigation of such processes "within the context of relatively gentle

upward air motion" (Houze, 1997). Stratiform precipitation accounts for (>85%) 73% of the area covered by rain and (>77%) 40% of the total rain amount across the (Mid-Latitudes) Tropics (D. Watters, personal communications; Schumacher and Houze (2003)). Stratiform rain can be well identified in radar data displays by a bright band, i.e. a pronounced layer of enhanced reflectivity corresponding to the melting layer (Fabry and Zawadzki, 1995).

In the past decade, several remote-sensing studies characterized micro-physical processes occurring in the ice (e.g. Kneifel

et al. (2011, 2015); Kalesse et al. (2016); Leinonen and Moisseev (2015); Oue et al. (2015b); Stein et al. (2015); Mason et al. (2018); Tridon et al. (2019)) and rain part of clouds (e.g. Williams (2016); Tridon et al. (2017b)). The commonality of all these studies resides in exploiting ground-based active (radar and lidar) and passive (microwave radiometer) instruments in a synergistic manner, with multi-frequency and/or Doppler and/or polarimetric radars constituting the backbone of the observing system. Multi-frequency methods (Battaglia et al., 2020a) rely on the fact that, when the wavelength of the radars becomes

comparable to the size of the particles being probed ("non-Rayleigh" regime), the measured reflectivity changes (typically decreases) relative to the Rayleigh regime, because the backscattered waves from different parts of the scatterer interfere (in a typically destructive way) with one another. Previous studies have demonstrated that dual and triple-frequency radar observations can provide additional information on bulk density and the characteristic size of the ice PSD (Kneifel et al., 2015; Battaglia et al., 2020b). Doppler (full spectral) information allow separating particles with different terminal velocities.

While this information is more valuable in rain than in ice, Doppler spectra allow for example to detect the presence of riming, which leads to an acceleration of the particle fall velocities above the typical $1\,\mathrm{ms^{-1}}$ observed for snow aggregate (Kneifel and Moisseev, 2020). The increasing terminal velocity of rimed particles causes the spectra to be first skewed, and, at larger riming, to become bi-modal (Zawadzki et al., 2001; Kalesse et al., 2016; Vogel and Fabry, 2018). Polarimetric radar observations are particularly sensitive to depositional growth in temperature regions which favour growth of asymmetric particle shapes (e.g.

needles, plates, dendrites). Observations obtained at the North Slope of Alaska (NSA) Atmospheric Radiation Measurement




(ARM) site have shown large signatures of differential reflectivity $Z_{DR}$ from plate-like crystals (Oue et al. (2015a)) whilst analysis of linear depolarization ratio LDR in the spectral domain enabled the identification of columnar ice crystal growth originated in liquid-cloud layers through secondary ice production (Oue et al., 2015b).

Whilst several studies have looked at the microphysical processes occurring within the melting layer (Drummond et al., 1996) and at the link between microphysical processes in snow above the freezing level and within the melting layer (Zawadzki et al. (2005); Li et al. (2020) and references therein), less attention has been paid to the analysis of quantitative relationships between ice microphysics just above the freezing level and rain microphysics just below the melting layer. This investigation can contribute to a holistic understanding of the chain of processes occurring in the cloud that lead to precipitation at the ground, which is key for model development but which may also help in better constraining full-column remote sensing retrievals, e.g. those applicable to space-borne radars like GPM, CloudSat and EarthCARE (Battaglia et al., 2020a) but also for improving QPE from ground-based radar observations (Gatlin et al., 2018).

A common assumption used across the melting layer is the conservation of water mass flux (e.g. Drummond et al. (1996)) which follows from assuming a stationary process and neglecting evaporation and condensation effects. The mass flux continuity assumption underpins several space-borne radar stratiform precipitation retrieval algorithms (e.g. Haynes et al. (2009); Mason et al. (2017)); in other retrievals where this constraint is not adopted, large discontinuities between mass fluxes above and just below the melting layer (Fig. 10 in Heymsfield et al. (2018)) are reported. This inconsistency between rain and snow mass fluxes pinpoints at the presence of some underlying issues in the snow retrievals, which are more uncertain (Heymsfield et al., 2018; Tridon et al., 2019).

In addition to water mass flux continuity several studies (Szyrmer and Zawadzki, 1999; Matrosov, 2008) further assume a one-to-one correspondence between the snowflake falling across the zero isotherm and the raindrop into which it melts (i.e. aggregation and breakup are neglected). We will refer to this as to the "melting-only steady-state" ("MOSS") assumption. Under this condition, there is a unique correspondence between DSD of raindrops and PSD of snowflakes. If true this property could indeed be used to constrain retrieval of hydrometeor vertical profiles in stratiform precipitation like done in Haynes et al. (2009) for the CloudSat space-borne radar.

The goal of this study is to propose a methodology applicable to multi-frequency Doppler polarimetric vertically pointing radar measurements which enables the investigation of the relationship between the microphysics of snow and of the rain produced via melting. Some of the science questions (SQ) that will be addressed in this paper are:

SQ1. What is the relationship between mass fluxes above and below the melting layer? How much does it deviate from the commonly-used constant mass flux assumption?

SQ2. Can information about rain microphysics and DSD (e.g. about the mean characteristic size) be used to better constrain the microphysical properties and PSD of the snow above?

SQ3. Are there specific ice cloud regimes (e.g. dominated by aggregation or riming) where the "MOSS" or the flux-continuity assumptions are more likely violated?





The paper is organized as follows: the dataset and the proposed methodology are presented in Sect. 2 and Sect. 3, respectively; Sect. 4 discusses the results for a case study in relation to the science questions; conclusions are drawn in Sect. 5.

## 2  Dataset

### 2.1  TRIPEx-pol field campaign

This study exploits the data collected during the "TRIple-frequency and Polarimetric radar Experiment for improving process observation of winter precipitation" (TRIPEx-pol). The campaign was conducted at the Jülich Observatory for Cloud Evolution Core Facility, Germany (JOYCE-CF 50° 54' 31''N, 6° 24' 49'' E, 111 m above mean sea level, see Löhnert et al., 2015) from November 2018 until February 2019. JOYCE-CF is a triple-frequency radar site (Dias Neto et al., 2019) including permanent installations of a X-, Ka-, and W-band vertically pointing Doppler radars. The quality of the remote measurements is continuously monitored with a number of auxiliary sensors, including a Pluvio rain gauge, Parsivel optical disdrometer (Löffler-Mang and Joss, 2000), microwave radiometers, and a Doppler wind lidar installed nearby the radars. In order to maximize radar volume matching, all the three radars are installed on the same roof platform within 10 m (see Tab. 1 for the technical specifications of the radars). Due to differences in the integration time of the radars and differences in the antenna beam widths, the data was averaged over 6 s in order to at least partially compensate for these factors. Because differences in the range resolution do not exceed 20% and are difficult to correct for, the data at W and X-bands are simply re-sampled at the Ka-band range bin resolution.

**Table 1.** Technical specifications and settings of the three vertically pointing radars operated during TRIPEx-pol at JOYCE-CF. Note that the W Band radar is a FMCW system for which chirp repetition frequency, number of spectral average, Nyquist velocity and range resolution are changing for different range intervals (see details in Dias Neto et al. (2019)); values provided here are for the lowest range gate region from 220 to 1480 m. Additionally, the radome of the W-Band is equipped with a strong blower system which avoids rain from accumulating.

| Specifications | X Band | Ka Band | W Band |
|---|---|---|---|
| Frequency [GHz] | 9.4 | 35.5 | 94.0 |
| Pulse Repetition Frequency [kHz] | 10 | 5.0 | 6.6 |
| Number of Spectral Bins | 4048 | 512 | 512 |
| Number of Spectral Average | 10 | 20 | 13 |
| 3dB Beam Width [°] | 1.0 | 0.6 | 0.5 |
| Sensitivity at 1km [dBZ], 2s integration | -50 | -70 | -58 |
| Nyquist Velocity [$\pm$ ms$^{-1}$] | 80 | 10.5 | 10.2 |
| Range Resolution [m] | 30 | 36 | 36 |
| Temporal Sampling [s] | 2 | 2 | 3 |
| Lowest clutter-free range [m] | 300 | 400 | 300 |
| Radome | No | No | Yes |



The absolute pointing accuracy of the scanning Ka-band radar has been estimated to be better than $\pm 0.1°$ in elevation and azimuth using a sun-tracking method. Following methodology of Kneifel et al. (2016), the mean Doppler velocity of the X and W-band radars have been compared to the Ka-band system for several cases with different horizontal wind velocities and directions. This analysis showed that the relative mis-alignment of the three radars is in the range of $0.1°$ which is expected to ensure a very high quality of the multi-frequency measurements.

## 2.2   The 24[th] November 2018 case study

The focus of our analysis is on a short time period (6:00–12:00 UTC) during a rain event on 24[th] November 2018. Selected radar measurements for this event are depicted in Fig. 1. The top and bottom of the melting layer have been derived with Linear Depolarization Ratio (LDR) from the Ka-band radar following the method described in Devisetty et al. (2019). Over the presented time period, the altitude of the zero degree isotherm was very stable and decreased by only 300 m from 1.1 km

at 6:00 to 0.8 km at 12:00. Radar reflectivity data below the bright band indicate two intervals of intensified rainfall: the first period is from 6:45 to 7:45 with the peak at 7:30, and a shorter interval that occurs around 9:00. Although for both periods similar X-band reflectivities are measured close to the ground (approx. 27 dBZ), the reflectivity and the Dual Frequency Ratio (DFR) data suggest completely different ice microphysics aloft. The first period is characterised by larger X-band echos in the ice part coinciding with extremely large $DFR^{X-Ka}$ values reaching 16 dB, which is a signature of strong aggregation and

presence of very large snowflakes (Kneifel et al., 2015). Almost no $DFR$ is measured after 7:45 which indicates relatively small ice particles. Note that the DFR data in ice were corrected for attenuation prior to the analysis. The attenuation due to the rain was derived from the Rayleigh part of the dual-frequency spectral ratio (see e.g. Tridon et al., 2013) assuming negligible attenuation at the X-band. The extinction due to melting particles was estimated from the rainfall rates retrieved below the melting layer with the methodology of Matrosov (2008). This technique has been shown to be in agreement with

multi-frequency Doppler spectra estimates (Li and Moisseev, 2019). These two components were added together and were used as a path integrated attenuation correction factor that is applied to the column. This methodology does not account for any attenuation within snow but this should be minimal at the X- and Ka- bands which seems to be confirmed by the fact that the DFR at the cloud top (Fig. 1), where Rayleigh targets are expected, is close to 0 dB.

    The Mean Doppler Velocity (MDV) is depicted in Fig. 1c. Despite high temporal variability of the Doppler data (the result

of vertical air motion and turbulence) a difference in dynamical properties of ice for the two periods is evident. MDVs of approximately 1-1.5 $\mathrm{ms}^{-1}$ in the first period are in agreement with simulations of large aggregates. Much larger velocities, especially after 8:00, suggest presence of rimed ice crystals (Kneifel and Moisseev, 2020).

    Fig. 1d shows the measured lidar backscattering crosssection from the celiometer that is located less than 5 m away from the radars. Thanks to these measurements, periods where the environmental conditions are favourable for riming can by identified.

Liquid clouds, that are essential for riming, appear as optically thick layers that strongly attenuate the light signal (Delanoë and Hogan, 2010; Van Tricht et al., 2014). The presented measurements exclude their presence before 8:00 for altitudes below 2 km. Afterwards, liquid clouds are detected in the vicinity or within the melting region. Unfortunately, due to strong attenuation no information about the presence of mixed phase clouds aloft is available.

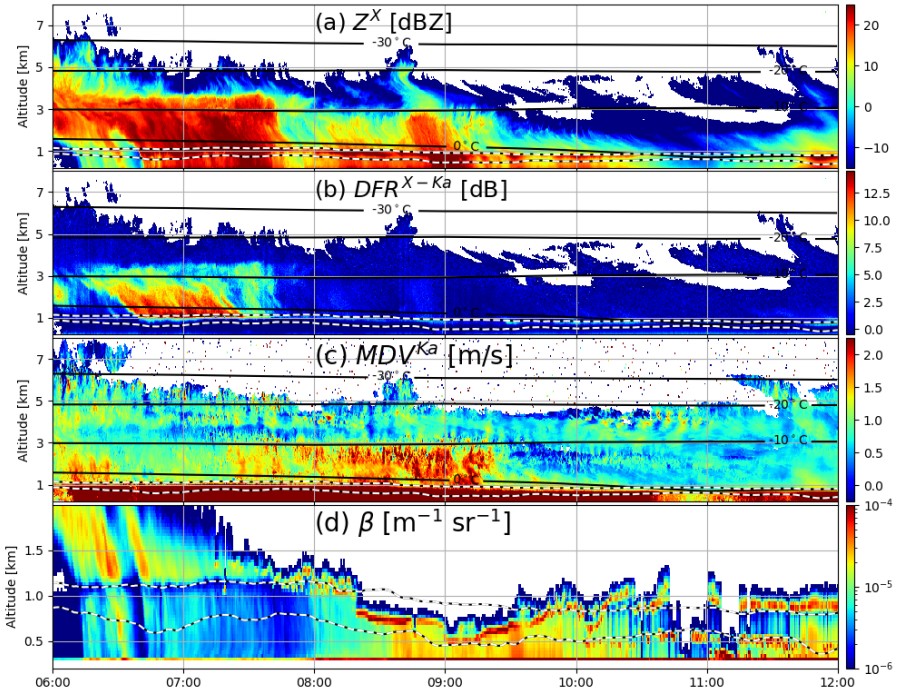

**Figure 1.** Time-height plots of radar variables measured at JOYCE on 24$^{th}$ November 2018: (a) X-Band radar reflectivity factor in dBZ; (b) Dual Frequency Ratio (DFR) of X and Ka bands ($Z^X - Z^{Ka}$); (c) Mean Doppler Velocity (MDV, Ka-band); (d) lidar backscattering crosssection (note the difference in the range of the presented altitudes). The dashed lines indicate the top and the bottom of the melting level derived from Ka-band Linear Depolarization Ratio (LDR). Black contour lines show isotherms derived from ECMWF analysis.

## 3   Methodology

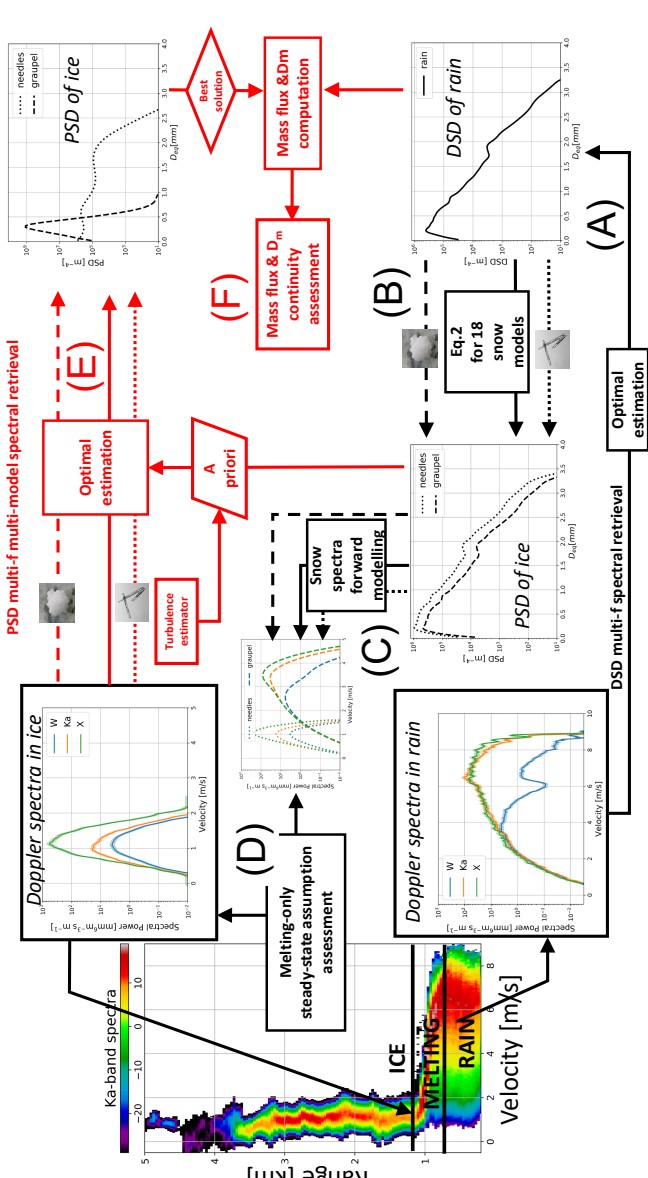

**Figure 2.** Schematic illustrating the rationale of the spectra closure study: by linking microphysical properties of rain just below the melting layer and of ice just above, the "MOSS" (black arrows) and the mass flux continuity (red arrows) assumptions can be evaluated.



This study aims at relating rain and ice microphysics immediately below and above the melting layer in stratiform precipitation. The overall logic of our approach is summarized in the schematic of Fig. 2. In a first approximation, retrieved rain properties can be exploited to infer information about the ice particles aloft via the "MOSS" assumption (follow back arrows). Rainfall properties can be derived with less uncertainty than for ice because terminal velocities and backscattering cross sections of raindrops are much more constrained than those of ice and snow particles. The predicted ice PSDs can then be used

to simulate radar snow spectra but only once a "snow model" is selected; the comparison between simulated and measured snow spectra allows to establish which snow models are more compatible with measurements and how realistic is the "MOSS" assumption. This bottom-up approach is not novel and has already been applied in the past (e.g. Drummond et al. (1996); Battaglia et al. (2003)).

  Here, thanks to the multi-frequency Doppler spectra approach, we can attempt a more elaborate "closure study" where more

accurate ice microphysical properties (and vertical wind) can be retrieved by matching spectra in ice at all frequencies via an optimal-estimation (OE) technique. For the apriori ice PSD, we use the PSD derived via the previous bottom-up approach based on the validity of the "MOSS" assumption. From ice PSDs and vertical wind, fluxes and PSD moments can be derived that can be directly compared to their counterparts in rain in a top-down approach, thus addressing the science questions (Sect. 1). Such procedure is featured in Fig. 2 with red coloured boxes and arrows. We now describe in detail the key steps of the whole

approach.

### 3.1   Rain DSD retrieval from multi-frequency radar Doppler spectra

Vertically pointing Doppler radars usually provide the full Doppler spectrum, i.e. the spectral distribution of the return power over the range of the line of sight velocities. Because the raindrop terminal velocity is an increasing and well constrained function of the raindrop size (Atlas et al., 1973), these measurements can be used to resolve the Drop Size Distribution (DSD),

once the vertical wind and turbulence are known (e.g., Williams et al., 2016; Tridon et al., 2017a; Giangrande et al., 2012).

  Our DSD retrieval in the range bins below the melting zone follows closely the steps described in Tridon and Battaglia (2015). The only modification here introduced, is the extension of the observation vector from 2 to 3 frequency bands in order to fully exploit the measurement capabilities of the radar site. The retrieval is based on Bayes' theorem (Rodgers, 2000): it minimizes the cost function that is composed of two equally weighted components. The first component computes the weighted

distance to the triple frequency spectra measurements, with the inverse variance of the measurement error used as a weight. The other term calculates the deviation from the prior knowledge of the DSD. For this retrieval, a widely adopted gamma-shaped DSD that fits the spectral measurements the best is used as a-priori estimate (for more detail see Tridon and Battaglia, 2015). The backscattering cross sections of raindrops are computed with a T-matrix method using the Python code of Leinonen (2014). The refractive index of water is computed at $10°C$ using a model of Turner et al. (2016). Terminal velocities of raindrops are

interpolated from a dataset of Gunn and Kinzer (1949) whereas the aspect ratio is calculated with a formula of Brandes et al. (2005). The orientation of raindrops is assumed to follow a normal distribution about $0°$ with standard deviation of $8°$ (Huang et al., 2008). Doppler spectra are simulated according to the methodology described in Tridon and Battaglia (2015) accounting

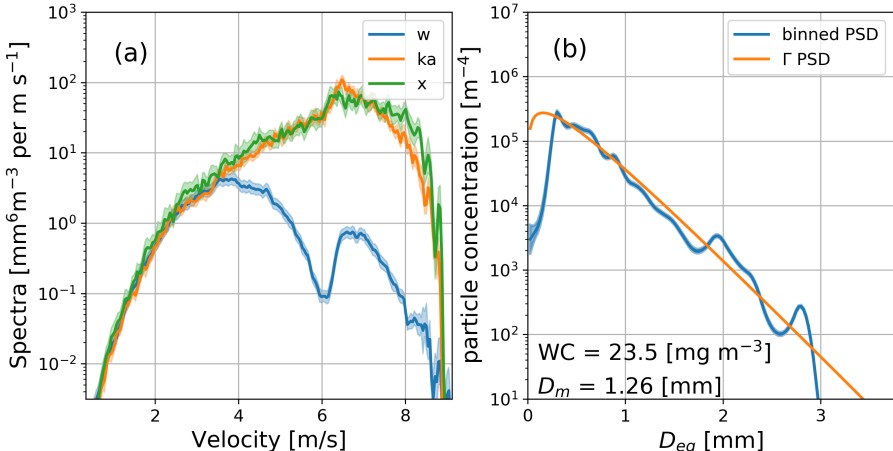

**Figure 3.** (a) Doppler spectra measured in rain just below the melting zone at 7:01:34 UTC. The green, orange and blue lines correspond to X-, Ka- and W-band data, respectively. The shaded areas represent the measurement uncertainties. (b) The corresponding binned DSD retrieval (blue) and the best fit gamma DSD (orange). The x-axis is the melted-equivalent diameter. The water content (WC) and mass weighted mean diameter ($D_m$) corresponding to the binned DSD are shown as a text in the corner.

for turbulence, vertical wind and radar noise level. The algorithm retrieves binned DSD along with two dynamical parameters: turbulence and vertical wind.

An example of the measurements and the corresponding retrieval is presented in Fig. 3. As expected, the spectral power for velocities below $4 \, \mathrm{ms}^{-1}$ is nearly identical for the different frequencies, which is a result of Rayleigh ($\propto D^6$) scattering at all bands for drops smaller than approximately $1 \, \mathrm{mm}$. This Rayleigh part of the spectrum can be used to determine differential path integrated attenuation for different radar bands (Tridon et al., 2013). The spectrally derived differential attenuation has been accounted for, prior to the retrieval. A significant reduction in the measured power at the W-band compared to the other frequency measurements can be found for velocities exceeding $4 \, \mathrm{ms}^{-1}$. This fall velocity regime corresponds to particle sizes for which non-Rayleigh scattering effects increase and culminate at the first resonant minimum, expected at $5.95 \, \mathrm{ms}^{-1}$ according to Gunn and Kinzer (1949) data. The difference between the measured and the anticipated position of the peak in the spectrum corresponds to the vertical air velocity (Kollias et al., 2002).

The blue line in Figure 3(b) shows the retrieved DSD that minimizes the cost function. This solution fits the measured radar reflectivity with an accuracy of $0.25 \, \mathrm{dB}$ at all frequency bands (not shown). As it can be seen, the widely used gamma model (orange line) represents well the bulk shape of the binned DSD for drops up to $3 \, \mathrm{mm}$ in size. Nevertheless, some subtle features of the Doppler spectra, such as an increase in the X- and Ka-band spectra around $6 \, \mathrm{ms}^{-1}$ that corresponds to a local DSD maximum around $2 \, \mathrm{mm}$, cannot be modeled with the gamma function. It should be noted that although we assume a gamma-shaped PSD as a prior, no explicit functional shape is assumed for the retrieved DSD. This is an important advantage of the spectral retrieval as it allows to retrieve complex DSDs, such as multi-modal distributions.



## 3.2 Deriving snow PSD from rain DSD via the "melting-only steady-state" assumption

In order to connect properties of ice with the properties of rain, several assumptions are made. Firstly, processes across the melting layer are assumed to be in steady-state. Secondly, effects of condensation or evaporation are neglected, which is supported by a very high relative humidity (RH) during the analysed period. The radiosonde launched at 9:00 UTC showed RH

values exceeding 90% in the proximity to the freezing level. Nevertheless, these measurements are likely to be underestimated because they were taken by the GRAW humidity sensor that often saturates around 90%. Moreover, as it is presented in Fig. 1d, signatures of liquid clouds were present in the lidar data for altitudes within the melting zone after 8:00 (Fig. 1d) that indicates water vapour supersaturation conditions after 8:15 UTC. These assumptions imply the flux of mass through the melting zone is conserved. Furthermore, following Szyrmer and Zawadzki (1999); Zawadzki et al. (2005); Matrosov (2008), no breakup and

no interaction between melting particles is assumed. Consequently, we might assume that one ice particle is converted into one raindrop and the mass of each particle is preserved through the melting layer $(m_s(D_s) = m_r(D_r))$; thus the particle number flux is conserved at any size. Mathematically:

$$N_s(D_s)[V_s(D_s) + w_s]\,\mathrm{d}D_s = N_r(D_r)[V_r(D_r) + w_r]\,\mathrm{d}D_r, \tag{1}$$

where $N_s$, $N_r$ denote the concentrations and $V_s$, $V_r$ are the still-air terminal velocities of snowflakes above the freezing level

(subscript s) and raindrops below the melting zone (subscript r), respectively. Vertical air motions $w_s$ and $w_r$ in snow and rain are assumed to be negative when upwards. Vertical air motions in stratiform precipitation can be assumed to be small compared to sedimentation velocities and hence they are neglected in the following. We will refer to this set of assumptions as the "melting-only steady-state" hypothesis. It is convenient to formulate Eq. 1 in terms of the equivalent melted diameter $D_{eq}$, which is a quantity preserved through the melting process:

$$N_s(D_{eq})V_s(D_{eq})\,\mathrm{d}D_{eq} = N_r(D_{eq})V_r(D_{eq})\,\mathrm{d}D_{eq} \qquad \Rightarrow \qquad N_s(D_{eq}) = N_r(D_{eq})\frac{V_r(D_{eq})}{V_s(D_{eq})}. \tag{2}$$

Equation (2) expresses a link between the PSD of ice and the DSD of rain resulting from melting of snow. It shows how the $v-D$ relationship for ice particles influences the shape of the underlying distribution of rain. This relation should be understood as a first order approximation that can be applied only when (1) the process is steady-state, (2) collision, coalescence and breakup are negligible, and (3) the relative humidity is close to the saturation level.

To verify if or where the "MOSS" assumption holds, the procedure shown in Fig. 2 as the black arrows is applied. First, the triple frequency measurements are extracted from the ranges just below the melting zone. Then, the full Doppler spectra are used to retrieve a binned rain PSD (step A). By applying the "MOSS" assumption through the melting zone, the rain DSD is mapped into the PSD of ice [step B, Eq. (2)]. The procedure is applied to 18 different snow models, described in detail hereafter in Sect. 3.3. For ease of display only two models (needles and graupel) representative of two extremes are

illustrated in the insets of Fig. 2. The number concentration predicted for the ice particles just above the melting layer depends on the snow model due to differences between their aero-dynamical properties, e.g. the aggregate models are characterised by higher particle concentration than rimed particles (compare the dashed-dotted with the dashed line in Fig. 2, Panel "PSD of ice"). Doppler spectra corresponding to each snow model are derived with scattering and aero-dynamical models (step C). The





resulting simulated spectra at the three bands are compared with the actual measurements (step D). As a first closure attempt,

simulated radar reflectivities for ice (corrected for attenuation using the methodology described in Section 2.1) and Doppler

velocities are compared with the measurements.

### 3.3    Snow models and Doppler spectrum simulator

The Doppler spectrum measured by a vertically pointing radar transmitting at the wavelength $\lambda$ is given by:

$$S_\lambda(v) = A_\lambda \times (S_{\lambda,target} * T_{air})(v) \qquad (3)$$

where $A_\lambda$ is the two-way attenuation, $S_{\lambda,target}$ is the reflectivity spectrum due to scattering from radar targets, $T_{air}$ is the

air broadening kernel while $*$ denotes the convolution operator (for more detail see Doviak and Zrnic, 1993). The reflectivity

spectrum can be expressed in terms of the particle size distribution and the backscattering crosssection as:

$$S_{\lambda,target}(v-w) = \frac{\lambda^4}{\pi^5 |K|^2} N(D_{eq}) \sigma_\lambda(D_{eq}) \frac{dD_{eq}}{dV}. \qquad (4)$$

where $\sigma_\lambda$ gives the backscattering crosssection of a target of a given size and $|K|^2$ denotes its radar dielectric factor and $v$ is

its terminal velocity that is assumed to be dependent on the particle size. Note that the vertical wind only shifts the spectrum

while mainly turbulence and wind shear cause a broadening of the spectrum. In this study a wide gamut of "snow models" is

considered to account for the large variability of scattering and aero-dynamical properties of ice crystals. Each "snow model"

entails a mass-size and an area-size relationship and provides size-dependent backscattering and extinction cross sections and

fall-speeds.

240        Broadly speaking two snow classes are analysed in this study. The first class consists of unrimed aggregates of different ice

habits i.e. needles, plates, columns, dendrites. These aggregates were created using the aggregation code described in detail in

Leinonen and Moisseev (2015). In total, approximately 30500 aggregates were generated. The total number of monomers, as

well as their size distribution have been varied, in order to produce a large variety of shapes and densities. The monomers are

distributed according to an inverse exponential size distribution, with the characteristic size ranging from 0.2 to 1 mm with a

minimum and maximum monomer size of 0.1 and 3 mm, respectively. The final aggregates consist of up to 1000 monomers,

and reach sizes of 2 cm. The scattering properties were obtained with the self-similar Rayleigh-Gans approximation (SSRGA)

(Hogan and Westbrook, 2014; Hogan et al., 2017). The SSRGA allows to approximate the scattering properties of an ensemble

of self-similar, low-density particles (such as aggregates) with an analytical expression and a set of corresponding fitting

parameters which characterize the structural properties of the simulated snowflakes. The full list of the SSRGA coefficients is

available at:https://doi.org/10.5281/zenodo.3746261.

        The second considered class contains snow particles generated by Leinonen and Szyrmer (2015); Leinonen et al. (2017)

comprised of aggregates of dendrites with different degrees of riming. Three riming scenarios are included in this dataset:

particles, which grew by riming only (model C, LS15C), aggregation and riming occurring simultaneously (model A, LS15A),

or subsequently (model B, LS15B). The degree of riming is expressed in terms of the equivalent liquid water path ranging from





| snow model | $\alpha$ [ms$^{-1}$] | $\beta$ [ms$^{-1}$] | $\gamma$ [m$^{-1}$] |
|---|---|---|---|
| plate | 1.41 | 1.43 | 1330.30 |
| dendrite | 0.89 | 0.90 | 1475.10 |
| column | 1.58 | 1.60 | 1552.29 |
| needle | 1.08 | 1.09 | 1781.26 |
| col. & dend. | 0.93 | 0.92 | 3628.93 |
| LS15A0.0 | 0.88 | 0.88 | 1626.17 |
| LS15A0.1 | 2.16 | 2.16 | 660.76 |
| LS15A0.2 | 2.09 | 2.09 | 936.83 |
| LS15A0.5 | 2.43 | 2.43 | 1400.49 |
| LS15A1.0 | 3.06 | 3.06 | 1199.37 |
| LS15A2.0 | 3.96 | 3.96 | 860.00 |
| LS15B0.1 | 1.25 | 1.25 | 1874.71 |
| LS15B0.2 | 1.53 | 1.53 | 2144.21 |
| LS15B0.5 | 2.29 | 2.29 | 1707.05 |
| LS15B1.0 | 3.25 | 3.25 | 1161.20 |
| LS15B2.0 | 4.59 | 4.59 | 715.88 |
| LS15C | 6.03 | 6.03 | 443.07 |

**Table 2.** Coefficients of the Atlas-like velocity-size relation (Eq. (5)) for different snow models.

0 kgm$^{-2}$ for dry aggregates to 2 kgm$^{-2}$ for graupel like particles. For instance, "LS15A1.0" denotes the model of aggregates grown by simultaneous riming and aggregation, where particles passed through a layer of 1 kgm$^{-2}$ of cloud droplets.

The terminal velocities of individual particles in the two snow classes are simulated for a standard atmosphere using the methodology of Böhm (1992). Then the expected velocity-size formula for each snow model is generated by a least square difference fit of the generated data to the Atlas-like formula (Atlas et al., 1973) (suggested by Seifert et al. (2014) as applicable
to snow as well):

$$V(D_{eq}) = \alpha - \beta \exp(-\gamma D_{eq}), \tag{5}$$

where $\alpha$, $\beta$, $\gamma$ are the optimal fitting parameters. The shape of this fitting function is more realistic than the frequently used power law fits since it can reproduce velocity saturation at larger sizes. Moreover, this parameterization is characterised by considerably smaller root-mean-square-error of the fit than the traditional power law approach. A complete list of the fitting
parameters corresponding to the different snow models is given in Tab. 2.





### 3.4 Optimal estimation retrieval of ice PSD based on multi-frequency Doppler spectra

While the bottom-up approach (comparing measured and simulated ice Doppler spectra based on rain DSD) only provides a qualitative evaluation of the "MOSS" assumption, we aim to directly derive the ice PSD from the measured multi-frequency Doppler spectra just above the freezing level. The principal of this OE retrieval is very similar to the OE retrieval used for

rain. Of course, the more complex scattering and terminal velocity behavior of snow must be accounted for and will also likely increase the retrieval uncertainties.

The main difference between the OE retrieval in rain and ice is the estimate of the prior state vector. In rain, we used the gamma model DSD that best fits the spectra (Tridon and Battaglia, 2015); in ice, the PSD that is predicted from the rain below the melting band assuming the validity of the "MOSS" assumption and following the methodology described in

Sect. 3.2, is used. An uncertainty of a factor of two is assumed for the prior PSD concentration. A prior for turbulence is derived using the method proposed by Borque et al. (2016). The velocity of the slowest detectable radar targets in ice is used as prior for the vertical velocity (step E). The uncertainties of these estimates are set to 175% in case of the turbulence and $0.16 \ \mathrm{ms}^{-1}$ for the vertical wind. These uncertainty values are derived from the corresponding root mean square differences between the first guesses and the final estimates in rain over the analysis period. An additional constraint that aims at improving

consistency between the mass fluxes is imposed, i.e. the fractional difference between the fluxes in ice and rain weighted with an uncertainty of $1/3$ is added to the cost function. This term in the cost function is much smaller than the one corresponding to the measurements, therefore it is effective only when the radar observations can be fitted by a family of PSDs spanning a range of mass fluxes. The retrieval is tested for all the selected models independently, and the model that corresponds to the lowest cost function is chosen as the best solution. For the best matching solution, parameters like mass flux and equivalent

mass-weighted size can be computed and directly compared with the same parameters in the rain (step F). This allows to achieve the "closure" and, for instance, to assess the validity of the flux continuity assumption.

## 4 Results

### 4.1 DSD retrieval

The goal of this study is to link the properties of rain with the characteristics of the overlying ice in stratiform precipitation.

As the rain DSD is the basis for this closure analysis, we first compare the spectra-retrieved DSD with the Parsivel2 measurements at the ground (Fig. 4). Despite the vertical distance of approximately 700-800 m between the disdrometer and the radar retrieved DSD just below the melting zone, the two methodologies provide comparable results. The comparison reveals several advantages of the radar-derived DSDs: first, it is able to retrieve smaller drops sizes ($D_{eq} < 0.5$ mm) that are not detected by the disdrometer (Thurai, 2019; Thurai and Bringi, 2018; Raupach and Berne, 2015); second, much higher temporal resolution

(6 versus 60 s); third, more reliable estimate of the number of large drops that are very infrequent and may not be captured by the limited sampling volume of the disdrometer. Note that the spectral method has also its limitations, e.g. the retrieval for $D_{eq} < 0.2$ mm must be interpreted with caution due to increasing uncertainties (see Tridon et al., 2017a).



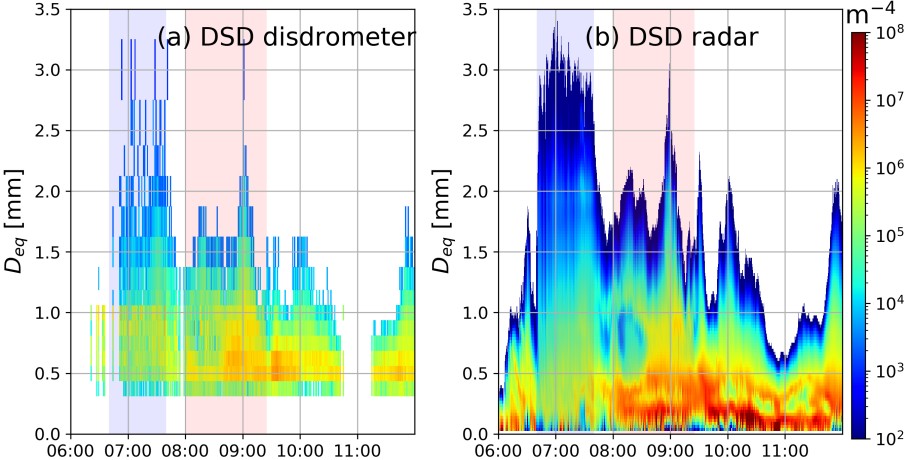

**Figure 4.** (a) DSD measurements at the ground with a Parsivel disdrometer. (b) DSD retrieved with multi-frequency Doppler spectra below the melting zone at ca. 700-800 m. The period shaded in blue correspond to the region of large $DFR^{X-Ka}$ above the melting layer that indicates aggregation. The period of enhanced Doppler velocities indicating riming is marked in red.

Throughout the rest of the paper, the period of large $DFR_{X-Ka}$ values (aggregation) is marked by the light blue color, whereas the domain of enhanced Doppler velocity (riming) is shaded in red. The DSDs during the two periods are quite distinct:
the aggregation dominated period (almost one hour) is associated with larger number of big drops and almost exponential DSDs. During the following riming period, much larger concentration of small droplets and multi-modalities of the DSD are found. There are two potential sources of this high concentration of small droplets: super-cooled drizzle that forms aloft by coalescence of supercooled cloud droplets or secondary ice crystals, e.g. generated by the Hallett-Mossop process (Mossop, 1976). In the first scenario, the slowly falling mode does not change significantly its intensity and position in the Doppler
spectrum while passing through the melting zone (Zawadzki et al., 2001) because there is no phase change of the particles. In the second scenario, the melting process changes the velocities and backscattering properties of the hydrometeors, thus resulting in a shift and a change in amplitude of the spectral power of the mode. The following analysis of the evolution of the Doppler spectra from the ice to the rain part are therefore expected to better explain the source of the small droplet mode.

### 4.2 Doppler spectral features during the investigated time periods

Differences between riming and aggregation regimes are reflected in the Doppler spectra that are shown in Fig. 5. During aggregation, the spectra in the ice phase are unimodal and the position of the peak is relatively constant at different heights, which indicates weak vertical air motion (Fig. 5a). The transition from ice to rain, corresponding to a strong broadening of the spectra, happens very rapidly within less than 200 m. The spectra from the riming period (Fig. 5b) are characterised by a much thicker melting layer (approx. 400 m) and by bimodal distributions both in rain and in ice. The secondary ice mode
appears approximately 1–1.5 km above the melting level, which corresponds to temperatures ranging between -4 and -6.5°C

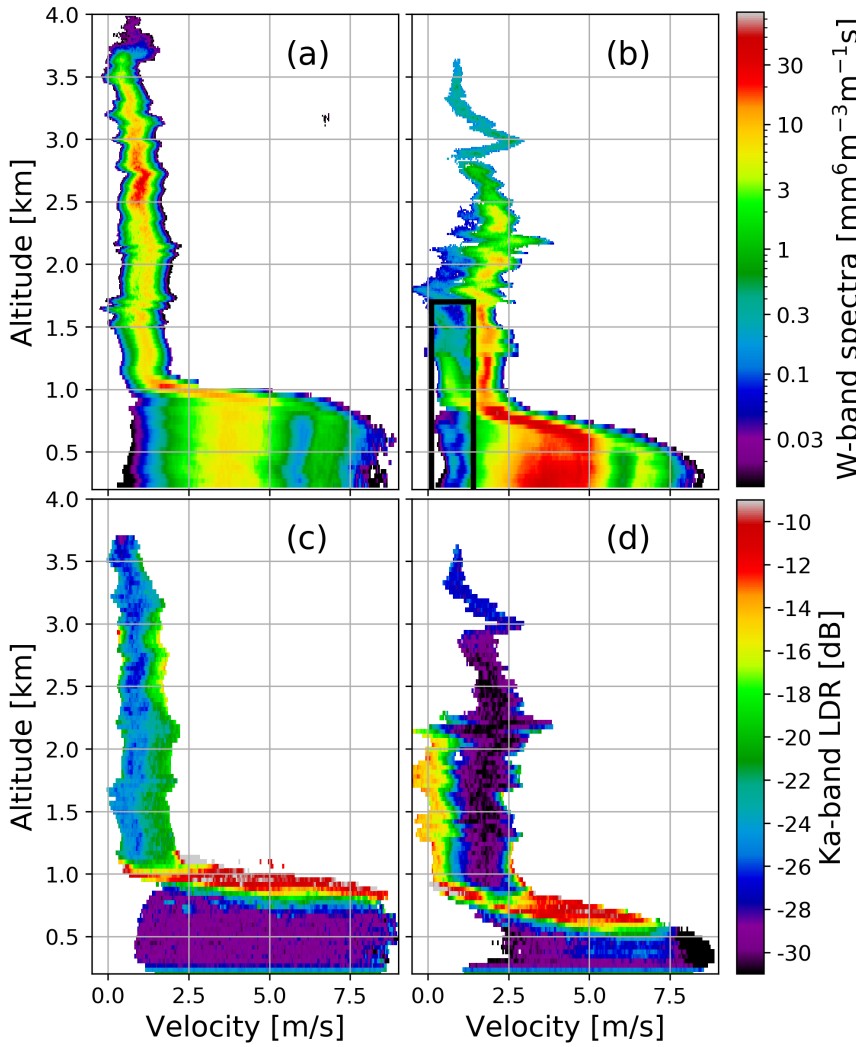

**Figure 5.** W-band Doppler spectra ((a) and (b)) and Ka-band spectral LDR ((c) and (d)). Panels (a) and (c) correspond to the measurements at 7:01 UTC dominated by aggregation; panels (b) and (d) were sampled at 8:58 UTC, when mean Doppler velocities indicate presence of rimed particles. Only the data where SNR> 3 dB are shown. The black box in panel (b) marks the secondary modes in ice and rain. Positive Doppler velocities indicate motions towards the radar.

according to the radiosonde launched at 9 UTC. There is high vertical variability in the position of the main peak, which indicates more dynamical conditions. The secondary mode increases its intensity while approaching the melting level but remains clearly separated from the main peak (see Fig. 5b). In the melting zone this separation disappears, the fall-speed of the secondary mode increases so that the secondary peak stretches out in the velocity domain and merges with the primary mode. This behaviour excludes the scenario of super-cooled drizzle above the freezing level as it was discussed before. The LDR measurements at the Ka-band (Fig. 5d) are in agreement with this theory. The slowly falling mode corresponds to LDR






reaching -15 dB; such values are much larger than those expected for nearly spherical drizzle droplets. This spectral feature is similar to the enhanced LDR signatures found in Oue et al. (2015b); Giangrande et al. (2016). They related the high LDR region to columnar ice crystals grown in liquid-cloud layers through secondary ice production. Interestingly, the high LDR
signature of the small ice mode can be also detected during the melting of these particles which might imply that the columnar crystals are of considerable size as they seem to maintain their asymmetric shape for quite some time until they are completely melted into drops (Fig. 5d). During aggregation, the opposite is true, i.e. the Ka-band LDR of large snowflakes is clearly larger than that of small ice crystals. This increase of LDR for large aggregates is principally consistent with scattering simulations of realistic snowflakes in Tyynelä et al. (2011) (their Fig. 7) where LDR values are predicted to increase for maximum sizes
exceeding 5 mm.

Note that the secondary mode in rain (Fig. 5b) appears to be disconnected from the secondary mode in ice during riming. At an altitude of approx. 800 m there is a clear gap between them, which is visible by the black box in Figure 5b. This separation is present over several minutes which suggests that the small rain droplets do not originate from the melting of ice crystals; thus the assumption of one-to-one correspondence between ice particles and rain drops may not hold for this profile. Lidar
measurements (Fig. 1d) indicate presence of a small droplets within the melting zone, therefore the secondary mode in rain is likely to be drizzle generated by this liquid layer or melting ice crystals (too little to be detected by the radar) that undergone rapid growth through collision-coalescence processes while passing through the cloud.

### 4.3 Inferring ice PSD based on rain DSD via the "melting-only steady-state" assumption

In a first step, we derive the DSD of ice from the PSD of rain via the "MOSS" assumption [Eq. (2)]. Fig. 6a shows the DSD
mass-weighted mean diameter ($D_m$) and the water content (WC), as it is retrieved from the Doppler spectra in the rain below the melting region. The aggregation period is characterised by smaller water content but larger characteristic size of rain drops compared to the riming period. With the "MOSS" assumption, the ice WC and the melted $D_m$ of snow depend on the $v - D$ relationship of ice and on the rain DSD. Because the velocities of raindrops are larger than those of the same-mass snowflakes of any density, it follows that $N_s(D_{eq}) > N_r(D_{eq})$. Consequently, the assumptions made in Section 3.2 imply that the ice
WCs at the freezing level are always larger than the rain WCs just below the melting zone (Fig. 6b). In the most extreme scenario, i.e. in case of slow dendrite aggregates, ice WC can be seven times larger than rain WC. For rimed snowflakes, this difference is much smaller, but still a factor of two is expected for graupel-like particles. Because the ratio $V_r(D_{eq})/V_s(D_{eq})$ is not constant but rather monotonically increasing with size, the ice PSD is not simply a scaled version of the underlying DSD of rain, i.e. the number concentration of large particles is reduced compared to the small ones. This causes a reduction of the
mean mass weighted diameter ($D_m$) during melting, i.e. the expected characteristic size of the DSD below the melting zone is up to 30% smaller than the corresponding size in the ice aloft (Fig. 6c) and this change is purely ascribed to aerodynamic effects combined with the mass flux conservation constraint. Note that for the majority of the particle models, the associated difference in $D_m$ usually does not exceed 10%; for rimed particles this change is even less pronounced and the characteristic melted-equivalent size is practically preserved.



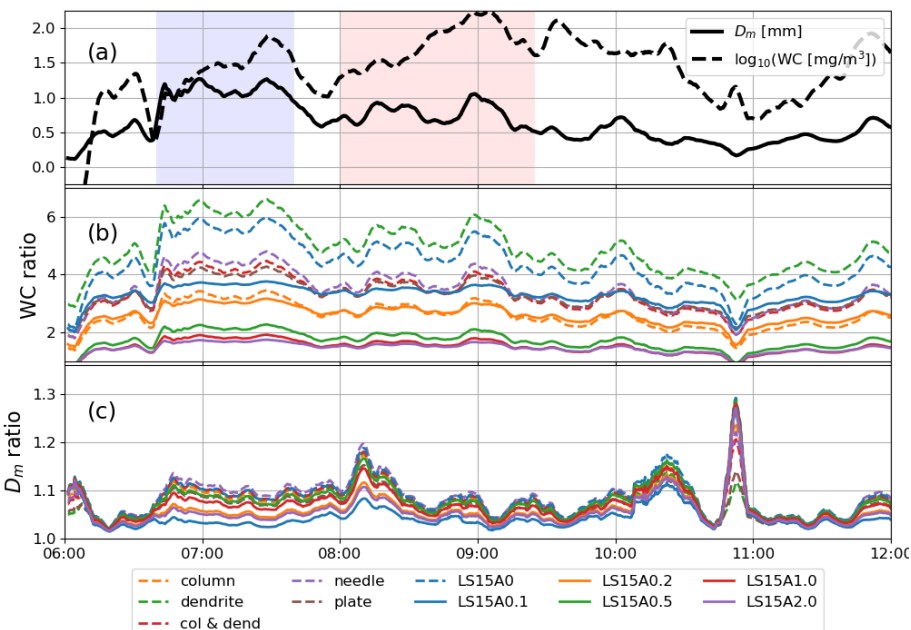

**Figure 6.** Panel (a): derived $D_m$ and WC (in log units) for the rain DSD just below the melting zone. Panel (b): relative change of the WC when passing from rain to ice, i.e. $WC^{ice}/WC^{rain}$. Panel (c): the same as (b) but for $D_m$, i.e. $D_m^{ice}/D_m^{rain}$. Different colors correspond to different ice models as indicated in the legend. Dashed lines correspond to unrimed aggregates, solid lines denote rimed particles. Blue/red shading indicates aggregation/riming dominated periods.

### 4.3.1 Discussion of the validity of the "melting-only steady state" assumption

According to the "reflectivity flux" method proposed by Drummond et al. (1996); Zawadzki et al. (2005), the ratio of reflectivity fluxes in snow and rain,

$$\gamma \equiv \frac{Z_s \, V_{D,\,s}}{Z_r \, V_{D,\,r}} \tag{6}$$

with the mean Doppler velocity $V_D$, is equal to $\mu \equiv (\rho_w/\rho_i)^2 \left(|K_i|^2/|K_w|^2\right) = 0.23$. The relation is only valid for Rayleigh
targets (which should hold for our X-band data) and under the "MOSS" assumption. Values between 0.15 and 0.30 are still compatible with the "MOSS" assumption when plausible vertical air motions (i.e. $w_r = \pm 1 \text{ms}^{-1}$ and $w_s = \pm 0.5 \text{ms}^{-1}$) are allowed (Drummond et al., 1996). If we introduce a normalized parameter in logarithmic units

$$\gamma_n[dB] = 10\log_{10}\left(\frac{\gamma}{0.23}\right), \tag{7}$$

values of $\gamma_n$ higher (lower) than 0 dB are indicative of break-up (collision-coalescence) or of a non stationary process. The
methodology has been recently applied to X-band profiler data by Gatlin et al. (2018) where it was found that thicker melting layers generally correspond to negative $\gamma_n$, i.e. are indicative of dominant coalescence and/or aggregation while transitioning



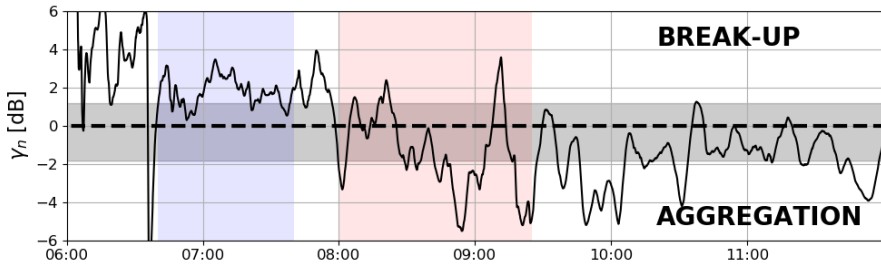

**Figure 7.** The normalised ratio between the reflectivity fluxes in ice and rain in vicinity of the melting level as defined by Eq. (7).The grey shading highlights the uncertainty introduced by the variability in the vertical wind.

from ice to liquid. Moreover, by combining (6) for $\gamma \equiv 0.23$ with (7), the ice reflectivity that would correspond to the "MOSS" assumption can be derived:

$$Z_{\gamma_n \equiv 0} = Z_m - \gamma_n, \tag{8}$$

where $Z_m$ is the reflectivity measured above the freezing level.

Most of the time, $\gamma_n$ is within the uncertainty limits introduced by vertical air motion (see Fig. 7). The root mean square difference over the case study between the ice reflectivity predicted with the "MOSS" hypothesis and the measurements is equal to 2.7 dB. The largest deviation is reported during the period when large snow aggregates are expected above the 0°C level. Large positive $\gamma_n$ values consistently suggest break-up as the main process occurring within the melting zone (Fig. 7).

The behaviour of $\gamma_n$ is more variable during the period of riming, where it oscillates between -6 and 4 dB. This non-uniform behavior can be, at least partially, caused by more turbulent environment, which might favour more non-stationary conditions. Also the presence of fallstreaks (e.g., around 9:00 UTC) can be seen as indication for more heterogeneous conditions. Moreover, riming particles have a broader range of terminal fall velocities (compared to aggregates of the same mass) which favours collision-coalescence processes and thus violating the underlying "MOSS" assumption. Within the uncertainty introduced by

the assumed vertical wind variability, our analysis confirms that the period before 8:00 is mainly characterised by break-up whereas the period after 8:00 UTC is dominated by collision-coalescence within the bright band. This corroborates the previous hypothesis of preponderance of aggregation before 8:00 and of riming after 8:00 within the snow layer.

### 4.3.2  Towards reconciling radar moments at the top of the melting layer by selecting adequate snow model

Encouraged by the results on the matching of the reflectivity fluxes in rain and ice, in the following section we test if the

"MOSS" assumption combined with the information on the DSD in rain can help in constraining microphysical properties of ice in vicinity of the melting level. For this purpose, the reflectivities at the three different frequencies are simulated for all the different snow models for the PSDs predicted with the "MOSS" assumption (Eq. 2). Regardless of the ice morphology the X-band reflectivity simulations cluster close together with a standard deviation between them ranging from 1 to 1.5 dB only (Fig. 8a). The envelope of simulations follows the X-band reflectivity predicted by assuming $\gamma_n = 0$ dB (denoted later as

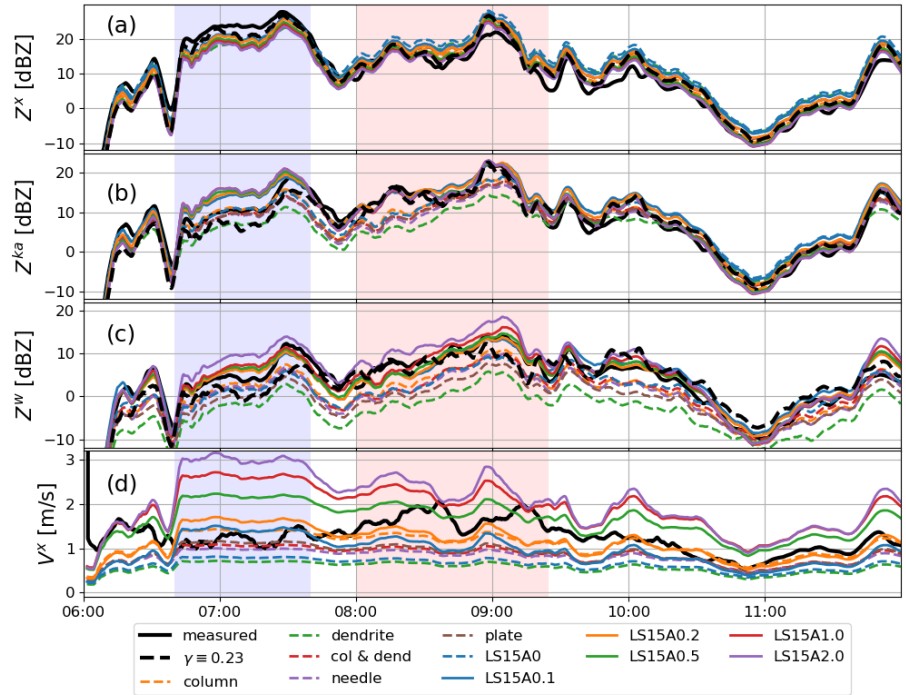

**Figure 8.** Measured (black lines) and simulated (coloured lines) radar reflectivities at the X-, Ka- and W-band and X-band MDV just above the melting level. The simulated values are predicted from the rain below the bright band by adopting the different snow models as listed in the legend. The black dashed lines show the reflectivity of snow derived with formula (6) for $\gamma \equiv 0.23$, i.e. the reflectivity corresponding to the "MOSS" hypothesis based on the radar reflectivity measured in rain and on MDV values in rain and ice (only X-band). Note, that the difference between the continuous and the dashed black line is equal to $\gamma_n$ (Eq. 7).

$Z^X_{\gamma_n \equiv 0}$) which is plotted as a dashed black line in Fig. 8. The largest difference in the simulated reflectivities occurs between the models of graupel (LS15C) and aggregates of dendrites; this discrepancy reflects differences in the ice water content for different snow models (see Fig. 6b) but is always smaller than 5 dB. The inter-model variability of the reflectivity simulations are comparable to the variability of the $\gamma_n$ which suggests that X-band radar data alone can provide very limited guidance on the density of snow above the melting zone, even when detailed information of the rain that originated from it is available.

The X-band simulations mirror the finding of the previous chapter, i.e. during the entire period of strong aggregation, the simulations underestimate the measurements by approximately 5 dB which is a signature of the "MOSS" assumption being invalid at that time period.

    The ranges of Ka and W-band simulated reflectivities is much wider than that at X-band with differences between heavily rimed particles and unrimed aggregates reaching 10 and 14 dB at Ka and W-band, respectively. This is related to the fact that

backscattering cross sections in non-Rayleigh scattering condition become increasingly sensitive to the snow particle type and density rather than simply being proportional to the square of the mass. Because the variability of simulated Ka- and W-band





reflectivities for different models is much larger than the range of $\gamma_n$ values, the triple-frequency reflectivity data are more informative about the particle models that are more suitable during specific time periods. For a qualitative comparison, $\gamma_n$ is used as a correction factor to the number concentration that makes triple-frequency measurements consistent with the "MOSS"

simulations. This is done via mapping the continuous into the dashed black line in Fig. 8 panels a-c ($Z^{Ka}_{\gamma_n \equiv 0} = Z^{Ka} - \gamma_n$; $Z^{W}_{\gamma_n \equiv 0} = Z^{W} - \gamma_n$). With this correction applied to the triple-frequency reflectivity data, it becomes clear that for the period before (after) 7:45, only models of unrimed aggregates (rimed particles) plotted with dashed (continuous) coloured lines are consistent with the multi-frequency observations. Similar conclusions are drawn when considering the simulated and observed Doppler velocities (Fig. 8d).

The $\gamma_n$-adjustment applied to all frequencies is a very crude approximation but it provides a significant improvement in terms of compatibility between triple frequency measured Doppler spectra moments. However it implies an "extensive" adjustment of the snow PSD; for instance a $\pm 3$ dB correction corresponds to doubling/halving the mass flux through the melting layer. Changes in the shape of the PSD could in principle lead to better fitting of the measurements and more continuous change in the mass flux. This is what is investigated next with the more exhaustive closure study.

## 4.4 Closure study: Connecting PSD and mass flux retrieved above and below the melting layer

Instead of only a qualitative comparison of the "MOSS" assumption (Step D in the schematic Fig. 2), we are now able to directly analyze the differences in mass flux and PSD above and below the melting layer by using the associated retrieval results for rain and ice. In this way, we can quantify the differences according to the dominating processes, which is expected to be also relevant for future modelling studies.

The PSD multi-spectral retrieval described in Sect. 3.4 (step E in Fig. 2) is applied to the whole period and the results are presented in Fig. 9. For each three minute time period, the mass flux and $D_m$ above the melting level (continuous lines in Figs. 9a-b) is derived with the snow model which has been found by the multi-frequency spectral OE to provide the optimal match with the observed spectra (Fig. 10). The best fitting snow models for the "aggregate" period are aggregates of needles or dendrites (note that LS15A0.0 represents unrimed aggregates of dendrites). During the period of enhanced Doppler velocities,

the retrieval suggests snow models of rimed particles. Rimed snow models also fit the measurements the best at later times, but because the characteristic size of particles is relatively low (Fig. 6), the mean Doppler velocity cannot be used to unequivocally confirm the presence of rimed particles. The agreement between the snow model selection discussed in Sect. 4.3.2 and the models suggested by the OE technique is quite remarkable which confirms a potential of using the "MOSS" hypothesis to at least reduce a number of plausible snow models in the analysis of the spectra just above the melting layer.

As a consistency check, the reflectivities for the derived PSD and snow model are compared in Fig. 9c, illustrating a very good match with the observations. The comparison of the derived mass fluxes in rain and ice (Fig. 9a) suggests, that for most of the time the mass flux across the melting zone is relatively well preserved. This is also reflected in the estimated total accumulations of rain (2.65 mm) and the melted equivalent accumulation of snow (2.60 mm) during the 6h time period, which shows only a 2% difference. As expected, there is a strong correlation (0.67) of the mass fluxes above and below the



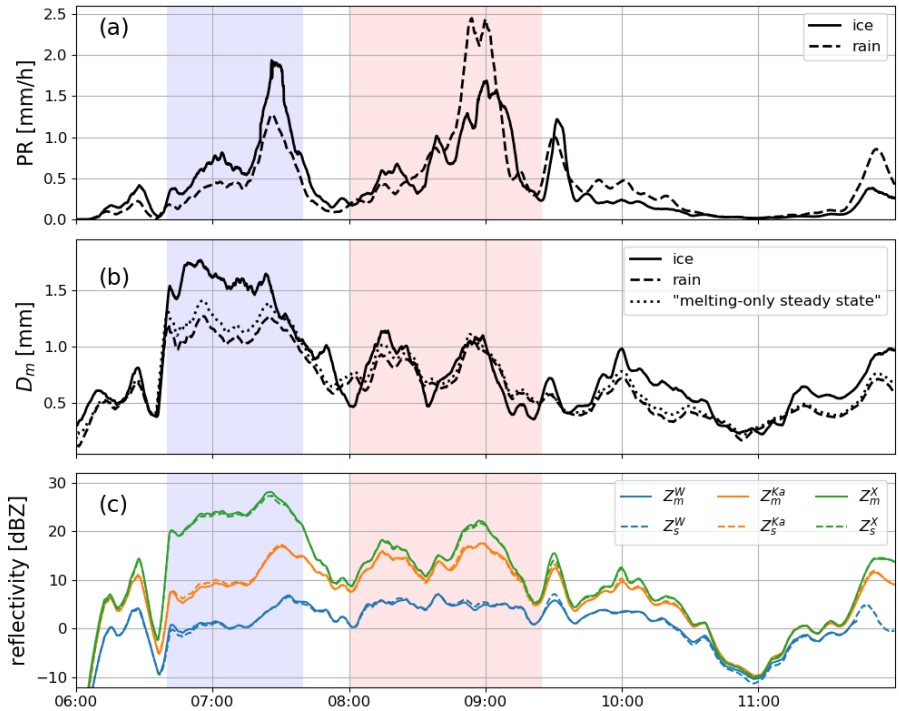

**Figure 9.** Results of the full Doppler spectra retrievals applied both below (Sect. 3.1) and above (Sect. 3.4) the melting zone. Panel (a): precipitation rate; panel (b): mean mass weighted diameter; panel (c): measured (subscript m) and simulated (subscript s) radar reflectivity values in the ice region just above the freezing level. The dotted line in panel (b) corresponds to the mass weighted mean melted-equivalent diameter of snow above the melting level that is predicted from rain with the "MOSS" assumption for the model that provides the best fit to the spectra measurements.

melting layer. Interestingly, the biggest differences in the mass flux are found for the maximum precipitation rates during the aggregation and riming period but with opposite sign.

During the aggregation period, the snowfall rate is on average 33% larger than the precipitation rate below the melting zone, which corresponds to a mean decrease from 0.82 to 0.55 mm/h. The characteristic size is found to be 23% larger in ice than one would expect, based on the "MOSS" assumption. Because aggregates are often composed of loosely connected crystals

(Garrett et al., 2012), this change in size could be caused by the break-up of the melting snowflakes as already conjectured in Sect. 4.3.1. The break-up hypothesis is also supported by a recent simulation study (Leinonen and von Lerber, 2018), where it was shown that the melting of the fragile ice connections within unrimed aggregates causes the particles to break into multiple droplets. Laboratory measurements of melting of snow aggregates, recorded under controlled temperature, relative humidity, and air velocity (Oraltay and Hallett, 2005), are also in agreement with this interpretation. However, the decrease

of precipitation rate during the aggregation period cannot be explained by break-up that doesn't affect the mass flux. Other





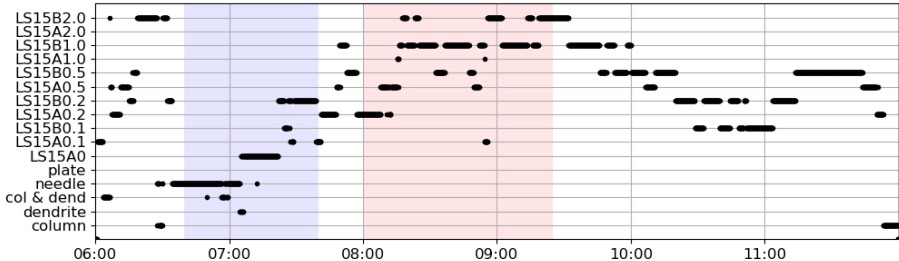

**Figure 10.** Snow models that match the best the full Doppler spectra measurements above the freezing level.

processes, such as evaporation and sublimation during melting would be needed to explain this reduction in the precipitation rate. If present, those processes would also contribute to a reduction of the characteristic size.

During the riming period the rain rate is approx. 15 % larger than the snowfall rate with the largest difference reported between 8:45 and 9:15 (approx. 28 % difference). This increase of precipitation rate could be explained by continuous riming
within the melting layer. The ceilometer data (see Fig. 1d) seem to indicate a layer of small liquid drops within the melting layer which might contribute to the enhanced rain rate by either riming or later also collision-coalescence of the small cloud droplets with rain drops from already melted snowflakes. However, the characteristic size appears not to change during this period, which is inconsistent with this hypothesis. Therefore, we speculate, that additional processes, such as shattering of large drops, might compensate for the raindrop size growth.

Considering the whole analysis period, we find the characteristic size of snow to be 21% larger than the one of the rain underneath. The root-mean-square-difference between them is equal to 0.23 mm only, whereas the correlation coefficient is 0.90. Our analysis shows, that with a simple scaling $D_m^{ice} = 1.21 D_m^{rain}$ the characteristic size of snow can be predicted with a root-mean-square-error of 0.12 mm. For the analysed case study this formula leads to slight underestimation during aggregation and to overestimation during riming. Recently published DWR statistics (Dias Neto et al., 2019) reveal that the very large DWR
signals found in this case study due to aggregation are relatively infrequent which suggests that even tighter relation between the hydrometeor sizes are expected for longer time series. If this tight relationship between snow and underlying rain can be confirmed for different locations and larger dataset, this would provide a very strong constrain for micro-physical retrievals (Tridon et al., 2019; Leinonen et al., 2018b).

## 5 Conclusions

This study investigates the link between rain and ice microphysics across the melting layer in stratiform rain. An OE technique applied to multi-frequency radar Doppler spectra is proposed in order to retrieve particle size distributions and dynamics both above and below the melting layer. This enables examining the variability of the precipitation rate and the mass-weighted melted diameter ($D_m$) across the melting region. The proposed technique is demonstrated for a 6h-long case study, observed





during the TRIPEx-pol field campaign at the Jülich Observatory for Cloud Evolution Core Facility and covering a gamut of ice

microphysical processes.

An initial assessment of the relationship between mass fluxes below and above the melting layer (scientific question 1) is based on the approach of Drummond et al. (1996) where the reflectivity fluxes (reflectivity × mean Doppler velocity) below and above the melting layer are compared. If the ratio of the two deviates from a value of 0.23 then the commonly adopted "melting only steady-state" ("MOSS") assumption is violated. During most of our case study, the reflectivity fluxes ratio is

within the uncertainty limits introduced by the vertical air motion (see Fig. 7) and the reflectivity fluxes are highly correlated (CC = 0.94). However, during the period of enhanced dual wavelength ratios ($DFR^{X-Ka} > 10$ dB), the reflectivity flux above the freezing level is systematically larger than in the underlying rain, indicating prevalent breakup during melting; therefore, the "MOSS" assumption seems to be breached, when large aggregates are present above the freezing level. More sophisticated analysis based on a comparison of binned PSDs retrieved in ice and rain from the triple frequency spectra measurements

is consistent with the findings based on the approach of Drummond et al. (1996); the mass flux across the melting layer is relatively well preserved (CC=0.67). The total accumulation of rain and snow differs only by 2% over the analysed case study. The largest difference between the fluxes above and below the melting level occurs during the aggregation and the most intense riming periods, with compensating effects. This analysis indicates that, not only the "MOSS" assumption but also the much weaker hypothesis of the mass-flux continuity across the melting zone is violated during the period of extreme aggregation.

In order to address the second science question related to linking characteristics of rain to the microphysical properties of ice aloft, the rain drop size distributions are retrieved below the melting level using the methodology of Tridon et al. (2017a). Then, the PSDs that would conserve the precipitation rate during melting are generated for all the analysed snow models by imposing "MOSS" assumption. Triple frequency Doppler spectra and their corresponding moments ($Z$ and $MDV$) in ice are simulated with the Self-Similar-Rayleigh-Gans approximation (Leinonen et al., 2018a). It is found that, at X-band, where snowflakes

behave mainly as Rayleigh scatterers, radar reflectivity of "MOSS"-generated PSDs of snow is only weakly dependent on the ice morphology. The standard deviation between the snow models is smaller than 1.5 dB and the difference between the most distinct models does not exceed 5 dB. This inter-model extend in the radar reflectivity simulations mainly reflects differences in the terminal velocities for different models. The range of reflectivity simulations is comparable to the uncertainty of the "MOSS" hypothesis (see range of values of $\gamma_n$ in Fig. 7) which suggests that the X-band data alone provides a very limited

guidance on the snow density. The range of Ka- and W-band reflectivity simulations for different snow models is much wider and reaches up to 10 and 14 dB respectively, which is related to the increasing dependence of backscattering cross sections on ice density and inner mass distribution at the higher frequency bands. It is found that the region where high MDVs are measured above the melting layer is clearly more compatible with the reflectivity simulations of rimed aggregates. The region of low MDVs and large $DFR^{X-Ka}$ can be better simulated with dry aggregate models. This indicates that when the high frequency

radar data are available, the "MOSS" assumption combined with the information on the drop size distribution (DSD) can guide selection of the snow models that represent bulk microphysics above the freezing level.

The analysis of the spectral retrievals in rain and ice reveals a strong dependence between the mean mass-weighted hydrometeor sizes for different phases. On average, the characteristic size of snow is 21% larger than the size of rain below and they



are highly correlated (CC=0.9). By using this linear dependence, the mean mass-weighted size of snow can be forecasted with an accuracy (RMSE) of 0.12 mm; nevertheless some underestimation (overestimation) is expected during periods dominated by aggregation (riming).

With respect to the third scientific question, whether there are specific ice cloud regimes where the MOSS assumption is more likely to be violated, we can only provide an answer for the relative short time period analyzed in this study. Regions dominated by aggregation above the melting layer tend to produce a reduction by approx. 33% in the flux and a decrease by 35% in the mean mass-weighted diameter when transitioning from ice to rain. In contrast, regions dominated by riming show an increase by approx. 15% in the flux and a relatively constant mean mass-weighted diameter. We hypothesize that the flux changes are associated to the variability of the relative humidity within the melting layer, with the regions dominated by riming more likely to be supersaturated as confirmed by the presence of a cloud layer. Ideally, measurements with differential absorption radar systems capable of characterizing in-cloud water vapor like those proposed in Battaglia and Kollias (2019); Roy et al. (2020) could assist in the interpretation of this kind of ground-based observations. On the other hand, the change of $D_m$ could be related to an increased likelihood for large aggregates to preponderantly undergo break-up in the melting zone. This is in agreement with theoretical (Leinonen and von Lerber, 2018) and laboratory (Oraltay and Hallett, 2005) studies which report breakup due to melting of the fragile ice connections within aggregates.

Our methodology should be applied to long-term observations in order to produce statistically significant results. Relationships between fluxes and characteristic sizes in ice and rain in stratiform precipitation are of great relevance since they can be directly used to constrain retrieval algorithms like those currently implemented in the framework of the Global Precipitation Measuring mission or envisaged for the EarthCARE mission. Uncertainties related to snow scattering models remain an obstacle in the accurate quantification of the ice phase microphysics. The integration of the findings of this study in a full-column rain-snow micro-physical retrieval of stratiform precipitation can pave the way towards a more refined selection of the snow model in line with the predominant ice microphysical process, thus advancing the current approach based on a single snow model assumption (e.g. Liao et al. (2016); Seto et al. (2013) for GPM). The proposed approach should help in better characterisation of the ice and rain microphysics just above and just below the melting layer, which will also highly benefit modelling studies of the processes occurring in the melting zone, which remain highly uncertain. Moreover, statistics on riming frequency would advance our knowledge of stratiform precipitation processes and lead to improvements in numerical weather models.

*Data availability.* All data obtained at JOYCE-CF are freely available on request from http://cpex-lab.de/cpex-lab/EN/Home/JOYCE-CF/ JOYCE-CF_node.html. Scattering tables are available at: 10.5281/zenodo.3746261

*Author contributions.* KM developed the algorithm. KM and AB drafted the paper. SK contributed to the scientific discussion. LT and MK developed snow aggregates scattering models. DO and MK derived sedimentation velocities of snow aggregates. All authors took part in editing the paper.



*Competing interests.*  The authors declare that they have no conflict of interest.

*Acknowledgements.*  The work by A. Battaglia was funded by the ESA-project "Raincast" contract: 4000125959/18/NL/NA. Work provided by S. Kneifel, M. Karrer, and D. Ori was funded by the German Research Foundation (DFG) under grant KN 1112/2-1 as part of the Emmy-Noether Group OPTIMIce. The radar and disdrometer dataset analyzed in this study were obtained at the JOYCE Core Facility (JOYCE-CF) co-funded by DFG under DFG research grant LO 901/7-1. The TRIPEx-pol campaign and work provided by L. von Terzi has
been supported by the DFG Priority Program SPP2115 "Fusion of Radar Polarimetry and Numerical Atmospheric Modelling Towards an Improved Understanding of Cloud and Precipitation Processes" (PROM) under grant PROM-IMPRINT (Project Number 408011764).



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
