# Peer review of "Linking rain into ice microphysics across the melting layer in stratiform rain: a closure study"

_Atmospheric Measurement Techniques, 2020_

## Referee Comment (RC1) · Anonymous Referee #1 · 18 Aug 2020

Title: Linking rain into ice microphysics across the melting layer in stratiform rain: a closure study.

Authors: Mróz, Kamil Battaglia, Alessandro Kneifel, Stefan von Terzi, Leonie Karrer, Markus Ori, Davide

DOI: 10.5194/amt-2020-272

Decision: Accept with minor revisions

General Comments: This preprint uses multi-frequency radar data and forward scattering calculations to investigate the validity of the often-used assumption of one raindrop corresponding to one snowflake, or "melting-only steady-state" (MOSS). The approach is extremely innovative, well-described, and supported, and should be of significant

interest to the community given the ubiquity of the examined assumption and need for better microphysical insight in ice regions and the melting layer. The manuscript is very strong, with a logical and thorough flow, and is also quite well-written, with only minor corrections and clarifications needed. Pending the following comments and technical corrections, I believe the manuscript will be ready for publication.

Specific Comments:

Line 50: Please add just a brief statement about why it is more valuable in rain than ice for readers less familiar with Doppler spectra techniques.

Line 54: Please change "asymmetric" to "nonspherical".

Line 103: By "re-sampled" here, I assume the authors mean "interpolated" and not a true re-sampling process (e.g., bootstrapping)? If so, please clarify.

Line 113: Because it forms the basis for sampling "above" and "below" the ML, please provide just a very brief description of what this approach entails.

Lines 197-199: If the lidar data indicates supersaturation due to the inferred presence of liquid clouds and thus little risk of evaporation, should that not imply that condensation will occur on melting ice particles (assuming their surface temperatures remains near 0C during melting, or at least colder than the environment) and thus violate the assumption of mass conservation? (In addition to the collision/coalescence of said liquid cloud droplets). I certainly understand this is being presented as a known simplifying assumption (as stated on lines 213-214), but what is written (i.e., that the assumed presence of supersaturated conditions from the lidar data supports the notion of mass conservation in the ML) may not be strictly true.

Line 347-349: If the ratio Vr/Vs increases with size, and Ns (compared to Nr) scales with this ratio, shouldn't this result in a relatively larger number of large particles compared to small ones (compared to what is measured in rain), rather than the other way around? Such an understanding would also seem to correspond with the subsequent

statement of Dm decreasing during melting due to this shift.

Line 359: This sentence stating the mean Doppler velocity is equal to the adjustment factor for the dielectric constant between ice and rain confused me. Is this a typo, or a reference to the idea that the change in dielectric constant is often approximately offset by the change in fall velocity, as noted in Drummond? Please introduce the factor mu separately. Also, after reviewing Zawadzki et al. (2005) it is my impression, perhaps wrongly, that this relation is strictly only true if the density of snow is assumed to be independent of its size, otherwise a size-dependent value of |Ks| would be needed. If that is the case, it should be explained and added to the list of qualifications for when the relation is valid. In general, given the importance of this value and formulation, a bit more explanation of its origins may be helpful to readers.

Figure 7: Does this analysis account for the residence time of particles within the ML, or is it a direct one-to-one matching of above/below the ML at a single point in time? If so, could the authors state this and speak to what impacts, if any, trying to better account for this offset in matching time might have?

Figure 7: How was the terminal velocity of snow determined here given the different possible models? The retrieval of the dominant snow type is explored in the subsequent section, but it isn't clear to me if that was applied to this analysis or if something constant was assumed?

Line 373: By "largest deviation" do the authors mean most consistently large deviation, given the larger magnitude (in dB-space) dips during the riming period? Please clarify.

Line 405: What is meant here by "mapping the continuous into the dashed black line"?

Line 492: By "inter-model extend", are the authors referring to the range of simulated radar reflectivity values? Please clarify.

Technical Corrections:

Line 16: Remove comma after "that".

Line 36: I don't believe "mid-latitudes" or "tropics" needs to be capitalized.

Lines 37, 40-41, and elsewhere: Remove parentheses around reference year.

Lines 51, 176, Table 1, and elsewhere: Change "msˆ-1" to "m sˆ-1".

Line 77: Add "the" before "DSD" and "PSD".

Line 97: Remove "a" before "X-".

Line 106: Please add "the" before "methodology".

Line 255 and elsewhere: Change "kgmˆ-2" to "kg mˆ-2".

Line 336: Change "undergone" to "underwent"

Line 448, 509: Change "approx." to "approximately"

Line 460: Change "even tighter relation. . . are expected" to "an even tighter relation. . . is expected"

Line 462: Change "constrain" to "constraint"

---

## Referee Comment (RC2) · Anonymous Referee #2 · 30 Aug 2020

Overview:

The authors analyze data from ground-based triple-frequency radar, lidar, surface disdrometers and gauges, as well as environmental observations, with the aim of describing the vertical microphysical structure of liquid and ice-phase precipitation in a single, 6-hour period during which stratiform rain was observed. The data are of high quality and the methods involve optimal estimation of the precipitation PSD's using Doppler spectra and a collection of diverse particle models. The primary question posed is to what extent the flux of stratiform precipitation through the melting layer can be considered a steady, particle-mass-conserving process, and what microphysical mechanisms might lead to deviations from that kind of process?

The work is an original contribution, including the collected datasets which are fairly

unique. The data and methods are generally described quite well, with good clarity of language, and the figures appropriately demonstrate the points made in the manuscript. However, there are some questions on the interpretation of the data and especially the "closure" procedure that will require some substantial explanation and/or revision, as detailed in Major Points, below.

Major Points:

(1) Section 3.4: Up through section 3.3, the manuscript is of high technical quality, and the authors' approaches and interpretations appear mostly sound. However, section 3.4 describes the optimal estimation (OE) of ice-phase particle properties that utilizes the Doppler spectrum to estimate ice-phase particle PSD's for different assumed ice particle models, selecting the most appropriate particle model based upon which one minimizes the OE's cost function.

In and of itself, the OE is fine. The problem is that the OE is constrained by (a) initial guesses, or priors, of the ice particle PSD's supplied by the Doppler-spectrum-derived rain PSD's which are extended to ice using the "melting only steady state" (MOSS) assumption, as well as (b) a second objective function term that constrains the ice and rain mass fluxes to be closer (a difference fraction standard deviation of 0.33 is assumed). The MOSS assumption, in particular, is used to obtain a prior ice PSD that has the same mass flux as the rain below it. Clearly, the two prior terms (a) and (b), but especially (a), of the OE's objective function will tend to force the estimated mass fluxes of ice and rain to be more similar, regardless of the ice particle model chosen. But the primary purpose of the OE described in section 3.4 is to "assess the validity of the flux continuity assumption" as stated in the last sentence of that section. (The fact that the rain-spectrum-derived constraint is assigned a factor of two error doesn't really allow that much freedom to the OE solution, because as seen in Fig. 2b, e.g., a change of 1 x 10ˆ4 mˆ-4 to 2 x 10ˆ4 mˆ-4 in number density is not that large.)

Clearly, the application of such an OE could result in greater consistency of estimated

ice and rain mass fluxes, and so as formulated, the estimated ice-phase precipitation fluxes from this OE can't be used to independently evaluate how much consistency there is between ice and rain fluxes. But that is precisely what is done in section 4.4. Unless I'm missing something, this is circular reasoning and not a scientifically valid approach.

If the authors want to address the ice vs. rain flux continuity issue in a quantitative way, they would need to decouple their rain and ice estimation procedures: What if no priors (referenced in a and b, above) are included in the objective function described in section 3.4, or what if only some simple gamma-fit to the ice particle Doppler spectrum is used as a prior? Either would decouple the rain and ice-phase estimation. If some prior based on rain-related PSD's and the MOSS assumption is required to get a stable estimate of ice PSD's, then one must question the information content of the ice Doppler spectrum and whether there is any way of independently estimating the ice PSD's and mass fluxes directly from their Doppler spectra.

(2) p. 18, last paragraph of section 4.3.1, and p. 21 second paragraph: one of the difficulties of interpreting profile-type measurements is that one doesn't get a full 3D picture of the atmosphere, but just a 2D "curtain". Therefore, isn't it just possible that there was some horizontal variability of precipitation during the "aggregation" period and wind components perpendicular to the mean storm motion that could move aggregates of different concentrations into or out of the "curtain", so-to-speak? (At least evidence of vertical wind shear *within* the "curtain" is suggested by the tilted structures of Z and DFR in Figs. 1a and 1b, respectively.) The melting layer during the "aggregation" period had a depth of $\sim$400 m, and so if the particles fell with an average speed of $\sim$2 m/s, then they could potentially move laterally out of the 17 m wide radar beam in the $\sim$200 s it took them to fall through the melting layer. If the precipitation was not strictly horizontally homogeneous, then that could cause difficulties for the authors' microphysical interpretation.

My general point here is that particle breakup in the melting layer is not the only possible explanation for higher ice-phase reflectivity fluxes relative to rain reflectivity fluxes during the "aggregation" period. . ... all it would take is some horizontal variation of aggregates perpendicular to the "curtain" and some vertical variation of the horizontal wind.

Also, although breakup is certainly possible in the melting layer, melting aggregates could self-collect pretty efficiently as well.

Minor Points:

(3) Fig. 2 is a very informative reference, but some of the inset plots are very small and hard to read, particularly the snow spectrum panel above (C). Although these plots are meant to be symbolic, it would be good if they could be read more easily.

(4) p. 11, Eqs. (3) and (4), if v is meant to symbolize terminal velocity, shouldn't the capital V be used, as in Eqs. (1) and (2)? Also, I think w was previously defined in Eq. (1) as "negative upward". Shouldn't the w in Eq. (4) be similarly defined?

(5) p. 15, beginning of first paragraph: when comparing the "aggregate" ice spectra to the "rimed" ice spectra in Fig. 5 (a) and (b) it looks like both the "aggregate" and "rimed" have mean peaks that are pretty steady in velocity up to 1.75 km altitude. The "aggregates" have a deeper structure that is more consistent, while the "rimed" particles peter out above 1.75 km and the peak becomes variable. It's a very minor point, but I would say the "aggregates" have a consistent spectral peak to 4 km, while the "rimed" particles also show a vertically-coherent peak, but only up to 1.75 km.

---

## Author Comment (AC2) · 22 Oct 2020

Overview:

The authors analyze data from ground-based triple-frequency radar, lidar, surface disdrometers and gauges, as well as environmental observations, with the aim of describing the vertical microphysical structure of liquid and ice-phase precipitation in a single, 6-hour period during which stratiform rain was observed. The data are of high quality and the methods involve optimal estimation of the precipitation PSD's using Doppler spectra and a collection of diverse particle models. The primary question posed is to what extent the flux of stratiform precipitation through the melting layer can be considered a steady, particle-mass-conserving process, and what microphysical mechanisms might lead to deviations from that kind of process?

The work is an original contribution, including the collected datasets which are fairly unique. The data and methods are generally described quite well, with good clarity of language, and the figures appropriately demonstrate the points made in the manuscript. However, there are some questions on the interpretation of the data and especially the "closure" procedure that will require some substantial explanation and/or revision, as detailed in Major Points, below.

The authors would like to thank for all the comments and the suggested corrections. In particular, the comment on the constraining the snow retrieval with the properties of rain made us realize that the validation of the MOSS assumption is not truly independent and, in fact, it is biased toward the mass flux continuity constraint. We have tested a new version of the algorithm were these conditions are removed but we noticed that it leads to much higher uncertainties in the estimates of both the environmental parameters (wind and turbulence) and the PSD itself. Moreover, due to a large number of parameters involved in the retrieval, the algorithm is often trapped in a local minimum of the cost function and the spectra are not well fitted. Such local minima correspond to e.g. only left or right slopes of the spectra being fitted. Therefore, another methodology is currently tested where much lower number of the retrieval parameters is considered. For more detail please see the answers to the specific points.

Major Points:

(1) Section 3.4: Up through section 3.3, the manuscript is of high technical quality, and the authors' approaches and interpretations appear mostly sound. However, section 3.4 describes the optimal estimation (OE) of ice-phase particle properties that utilizes the Doppler spectrum to estimate ice-phase particle PSD's for different assumed ice particle models, selecting the most appropriate particle model based upon which one minimizes the OE's cost function.

In and of itself, the OE is fine. The problem is that the OE is constrained by (a) initial guesses, or priors, of the ice particle PSD's supplied by the Doppler-spectrum-derived rain PSD's which are extended to ice using the "melting only steady state" (MOSS) assumption, as well

as (b) a second objective function term that constrains the ice and rain mass fluxes to be closer (a difference fraction standard deviation of 0.33 is assumed). The MOSS assumption, in particular, is used to obtain a prior ice PSD that has the same mass flux as the rain below it. Clearly, the two prior terms (a) and (b), but especially (a), of the OE's objective function will tend to force the estimated mass fluxes of ice and rain to be more similar, regardless of the ice particle model chosen. But the primary purpose of the OE described in section 3.4 is to "assess the validity of the flux continuity assumption" as stated in the last sentence of that section. (The fact that the rain-spectrum-derived constraint is assigned a factor of two error doesn't really allow that much freedom to the OE solution, because as seen in Fig. 2b, e.g., a change of $1 \times 10^4$ m$^{-4}$ to $2 \times 10^4$ m$^{-4}$ in number density is not that large.)

Clearly, the application of such an OE could result in greater consistency of estimated ice and rain mass fluxes, and so as formulated, the estimated ice-phase precipitation fluxes from this OE can't be used to independently evaluate how much consistency there is between ice and rain fluxes. But that is precisely what is done in section 4.4. Unless I'm missing something, this is circular reasoning and not a scientifically valid approach.

If the authors want to address the ice vs. rain flux continuity issue in a quantitative way, they would need to decouple their rain and ice estimation procedures: What if no priors (referenced in a and b, above) are included in the objective function described in section 3.4, or what if only some simple gamma-fit to the ice particle Doppler spectrum is used as a prior? Either would decouple the rain and ice-phase estimation. If some prior based on rain-related PSD's and the MOSS assumption is required to get a stable estimate of ice PSD's, then one must question the information content of the ice Doppler spectrum and whether there is any way of independently estimating the ice PSD's and mass fluxes directly from their Doppler spectra.

The link between PSDs of ice and DSDs rain imposed in the retrieval was intended to stabilize the retrieval, but as you rightly pointed out, by using it the mass flux continuity conjecture cannot be evaluated independently. Therefore, we tested another version of the ice PSD retrieval where this dependence is removed. As expected, the retrieval became more uncertain which is manifested by much higher variability of the precipitation rate over time as it can be seen in Fig.1. below (left panel). As in the old analysis, the characteristic size of ice during the aggregation period is much larger than in the rain below which suggests break-up during melting. However, during the riming period, the precipitation rate in ice is on average larger than in rain which is not in line with our previous interpretation about the condensation and collision-coalescence with cloud droplets within the melting zone. An inspection of the individual spectra revealed two problems: first, the retrieval that is not constrained by the rain is often trapped in the local minima of the cost function that leads to poor spectra fitting. Second, Doppler spectra are often matched equally well by the two or more snow models by appropriately selecting the vertical wind velocity as it is shown in Fig.2. In order to address these issues, we are currently working on reducing the number of the ice PSD bins retrieved by the algorithm and constraining the wind component. The velocity of the vertical air motion is estimated from the velocity of the slowest detected targets. The final version of the retrieval will be included in the revised version of the manuscript. Figure 9 in the paper will show the spread of the best fitting solutions which will help in quantifying information content of the spectral data.

[Figure]

*Figure 1. The full Doppler spectra retrievals applied both below and above the melting zone. Left: no dependence between the properties of ice and rain is used; right: PSD of ice constrained by the MOSS assumption*

[Figure]

*Figure 2. The measured (continuous line) and the simulated spectra (dashed line) for two different snow models (left: L&S15A0.5, right: L&S15A2.0).*

(2) p. 18, last paragraph of section 4.3.1, and p. 21 second paragraph: one of the difficulties of interpreting profile-type measurements is that one doesn't get a full 3D picture of the atmosphere, but just a 2D "curtain". Therefore, isn't it just possible that there was some horizontal variability of precipitation during the "aggregation" period and wind components perpendicular to the mean storm motion that could move aggregates of different concentrations into or out of the "curtain", so-to-speak? (At least evidence of vertical wind shear *within* the "curtain" is suggested by the tilted structures of Z and DFR in Figs. 1a and 1b, respectively.) The melting layer during the "aggregation" period had a depth of ~400 m, and so if the particles fell with an average speed of ~2 m/s, then they could potentially move laterally out of the 17 m wide radar beam in the ~200 s it took them to fall through the melting layer. If the precipitation was not strictly horizontally homogeneous, then that could cause difficulties for the authors' microphysical interpretation.

My general point here is that particle breakup in the melting layer is not the only possible explanation for higher ice-phase reflectivity fluxes relative to rain reflectivity fluxes during the "aggregation" period... all it would take is some horizontal variation of aggregates perpendicular to the "curtain" and some vertical variation of the horizontal wind.

Also, although breakup is certainly possible in the melting layer, melting aggregates could self-collect pretty efficiently as well.

The authors fully agree with this comment. This aspect was omitted in the discussion because the presented analysis is based on the 1D assumption due to unavailability of the data about the spatial variability of the system. However, we acknowledge that raising this point is important in the interpretation of the data we present. Therefore, the following discussion was added at the end of section 4.3.1:

One of the difficulties of interpreting profile-type measurements is that they do not provide a full 3D picture of the atmosphere, but just a 2D slice. Therefore, the presented conclusions are based on the assumption that the observed system is locally homogeneous i.e. despite horizontal winds the measurements taken below the melting layer correspond to the evolution of the ice PSD measured aloft. Considering the horizontal wind speed within the bright band (approx. 1.8 m/s during the "aggregation" period according to the ECMWF model) and the time needed for the particles to melt (approx. 3 minutes based on the MDV data) the precipitating system must be uniform over 325 m to meet this criterion. Because, the beam with of the X-band radar at the altitude of the melting zone is only 15 m, it is possible that the higher ice-phase reflectivity flux relative to rain can be a result of a horizontal gradient of the reflectivity that, for example, corresponds to the storm intensification along the wind direction. Note that, for the most part of the aggregation period the precipitation rate increases over time (see Fig. 9a) which supports this alternative interpretation.

Minor Points:

(3) Fig. 2 is a very informative reference, but some of the inset plots are very small and hard to read, particularly the snow spectrum panel above (C). Although these plots are meant to be symbolic, it would be good if they could be read more easily.

Fonts of the labels was increased in the schematic that makes it much clearer.

(4) p. 11, Eqs. (3) and (4), if v is meant to symbolize terminal velocity, shouldn't the capital V be used, as in Eqs. (1) and (2)? Also, I think w was previously defined in Eq. (1) as "negative upward". Shouldn't the w in Eq. (4) be similarly defined?

The velocity symbol, v, in the equations (3) and (4) was capitalized. The vertical wind is consistently defined as "negative upwards". The formula (4) just indicates that whenever

non-zero wind is present the measured spectrum is shifted in the velocity domain. To avoid any confusion, an additional notation is introduced, i.e. $S_{\lambda,w,target}$ denotes the reflectivity spectrum affected by the vertical winds.

(5) p. 15, beginning of first paragraph: when comparing the "aggregate" ice spectra to the "rimed" ice spectra in Fig. 5 (a) and (b) it looks like both the "aggregate" and "rimed" have mean peaks that are pretty steady in velocity up to 1.75 km altitude. The "aggregates" have a deeper structure that is more consistent, while the "rimed" particles peter out above 1.75 km and the peak becomes variable. It's a very minor point, but I would say the "aggregates" have a consistent spectral peak to 4 km, while the "rimed" particles also show a vertically-coherent peak, but only up to 1.75 km.

The suggested comment was added:

The ``aggregates'' have a consistent spectral peak to 4 km, while the ``rimed'' particles also show a vertically-coherent peak, but only up to 1.75 km in altitude.

---

## Author Response (AR1)

Once more, on behalf of the co-authors, I would like to thank for all the comments and list of corrections that helped to improve the quality of our paper. The detailed responses to the raised issues are provided below. All the answers are written in red font to make them easier to find. I addition, a pdf file with all the modifications marked-up is combined with this document.

**Anonymous Referee #1**

Received and published: 18 August 2020

Title: Linking rain into ice microphysics across the melting layer in stratiform rain: a closure study.

Authors: Mróz, Kamil Battaglia, Alessandro Kneifel, Stefan von Terzi, Leonie Karrer, Markus Ori, Davide

DOI: 10.5194/amt-2020-272

Decision: Accept with minor revisions

General Comments: This preprint uses multi-frequency radar data and forward scattering calculations to investigate the validity of the often-used assumption of one raindrop corresponding to one snowflake, or "melting-only steady-state" (MOSS). The approach is extremely innovative, well-described, and supported, and should be of significant interest to the community given the ubiquity of the examined assumption and need for better microphysical insight in ice regions and the melting layer. The manuscript is very strong, with a logical and thorough flow, and is also quite well-written, with only minor corrections and clarifications needed. Pending the following comments and technical corrections, I believe the manuscript will be ready for publication.

Specific Comments:

Line 50: Please add just a brief statement about why it is more valuable in rain than ice for readers less familiar with Doppler spectra techniques.

The sentence was modified to accommodate some explanation: "While this information is more valuable in rain than in ice, since the velocity of raindrops is unambiguously related to their mass and size (which is not true of snow), Doppler spectra allow to detect the presence of riming..."

Line 54: Please change "asymmetric" to "nonspherical".

Changed

Line 103: By "re-sampled" here, I assume the authors mean "interpolated" and not a true re-sampling process (e.g., bootstrapping)? If so, please clarify.

**Thank you for pointing it out, it is changed to "interpolated"**

Line 113: Because it forms the basis for sampling "above" and "below" the ML, please provide just a very brief description of what this approach entails.

**The following sentence was added:**

"This approach is based on a very strong bright band signature in the LDR data in correspondence to the melting regardless of the rainfall intensity. In this study, the inflection points around the LDR peak are used as the top and the bottom of the melting zone."

Lines 197-199: If the lidar data indicates supersaturation due to the inferred presence of liquid clouds and thus little risk of evaporation, should that not imply that condensation will occur on melting ice particles (assuming their surface temperatures remains near OC during melting, or at least colder than the environment) and thus violate the assumption of mass conservation? (In addition to the collision/coalescence of said liquid cloud droplets). I certainly understand this is being presented as a known simplifying assumption (as stated on lines 213-214), but what is written (i.e., that the assumed presence of supersaturated conditions from the lidar data supports the notion of mass conservation in the ML) may not be strictly true.

Thank you for spotting this. Indeed, the discussion that was pointed out was not strictly true therefore it was modified to:

"In order to connect properties of ice with the properties of rain, several assumptions are made. Firstly, processes across the melting layer are assumed to be in steady-state. Secondly, effects of condensation or evaporation are neglected. The radiosonde launched at 9:00 UTC showed RH values exceeding 90% in the proximity to the freezing level which effectively excludes the possibility of evaporation. However, the possibility of condensation on melting ice particles and collision/coalescence with cloud droplets cannot be ruled out. Due to the saturation problem of the GRAW humidity sensor the RH measurements are likely to be underestimated which is confirmed by lidar measurements where signatures of liquid clouds are present within the melting zone after 8:15 UTC (Fig.1d) that indicates water vapour supersaturation conditions. Despite the potential inconsistencies of our assumptions with the actual state of the atmosphere, neglecting condensation and evaporation is used as a simplifying hypothesis that implies the flux of mass through the melting zone is conserved."

Line 347-349: If the ratio Vr/Vs increases with size, and Ns (compared to Nr) scales with this ratio, shouldn't this result in a relatively larger number of large particles compared to small ones (compared to what is measured in rain), rather than the other way around? Such an understanding would also seem to correspond with the subsequent statement of Dm decreasing during melting due to this shift.

That is true, the number concentration of large particles is increased compared to the small ones and it was modified in the text. Thank you for spotting this. We changed the word "reduced" to "increased in Line 360.

Line 359: This sentence stating the mean Doppler velocity is equal to the adjustment factor for the dielectric constant between ice and rain confused me. Is this a typo, or a reference to the idea that the change in dielectric constant is often approximately offset by the change in fall velocity, as noted in Drummond? Please introduce the factor mu separately. Also, after reviewing Zawadzki et al. (2005) it is my impression, perhaps wrongly, that this relation is strictly only true if the density of snow is assumed to be independent of its size, otherwise a size-dependent value of |Ks| would be needed. If that is the case, it should be explained and added to the list of qualifications for when the relation is valid. In general, given the importance of this value and formulation, a bit more explanation of its origins may be helpful to readers.

The sentence you are referring to is a typo. It has been corrected. We meant that the reflectivity flux ratio is equal to  $\mu = 0.23$ . Regarding the other point, the factor  $\mu$  in the study of Zawadzki et al. (2005) was derived assuming constant ice density. Nevertheless, their derivation is based on the formula of Debye that relates the dielectric constant to density of ice which effectively implies that the ice particles of the same mass, regardless of their density, correspond to the same reflectivity under Rayleigh scattering assumption. I add this comment in the manuscript:

"According to the ``reflectivity flux" method proposed by Drummond et al. (1996); Zawadzki et al. (2005), the ratio of the reflectivity fluxes in snow and rain,

**$\gamma = Z_s V_{D,s} / (Z_r V_{D,r})$**

is equal to  $\mu \equiv (\rho_w/\rho_i)^2 (|K_i|/|K_w|)^2 = 0.23$ , where the mean Doppler velocity is denoted by VD, and the subscripts *s* and *r* indicate sampling in snow and rain, respectively, whereas the subscript *i* indicates ice. The relation is only valid for Rayleigh targets (which should hold for our X-band data) and under the ``MOSS'' assumption. Although, the factor  $\mu$  was computed assuming constant ice density, the derivation is based on the formula of Debye  $(|K_s|/\rho_s = \text{const})$  which implies the reflectivity of the ice particles depends only on their mass not density. Therefore, the value of  $\mu$  is independent on the snow morphology."

Figure 7: Does this analysis account for the residence time of particles within the ML, or is it a direct one-to-one matching of above/below the ML at a single point in time? If so, could the authors state this and speak to what impacts, if any, trying to better account for this offset in matching time might have?

The following text is added in Line 387 to clarify it:

"In order to match the data below and above the melting layer more precisely, for each 15 minutes time window the optimal time lag that maximizes the correlation between the X-

band reflectivity in ice and rain is derived. All the results that follows use this optimal matching in time."

Figure 7: How was the terminal velocity of snow determined here given the different possible models? The retrieval of the dominant snow type is explored in the subsequent section, but it isn't clear to me if that was applied to this analysis or if something constant was assumed?

The analysis of Drummond et al. (1996) does not use snow models explicitly because the velocity of particles is assumed to be the one measured by the radar. Therefore, the measured mean Doppler velocity is used to derive the reflectivity fluxes in rain and ice (formula 7).

The following statement was added to make it clearer:

"Note that this method is based purely on the radar measurements thus it is not dependent on any snow or rain model"

Line 373: By "largest deviation" do the authors mean most consistently large deviation, given the larger magnitude (in dB-space) dips during the riming period? Please clarify.

Yes, we meant the consistent period, therefore we modified this sentence as follows: "The most consistent deviation from the uncertainty limits is reported during the period when large snow aggregates..."

Line 405: What is meant here by "mapping the continuous into the dashed black line"?

To avoid any confusion, we modified this sentence:

"For a qualitative comparison,  $\gamma_n$  is used as a correction factor to the number concentration that makes triple-frequency measurements consistent with the ``MOSS'' simulations. This is done by reducing the measured reflectivities by  $\gamma_n$  derived for the X-band ( $Z_{\gamma_n\equiv 0}^X = Z^X - \gamma_n$ ;  $Z_{\gamma_n\equiv 0}^{Ka} = Z^{Ka} - \gamma_n$ ;  $Z_{\gamma_n\equiv 0}^W = Z^W - \gamma_n$ ). The result of this correction is shown as the dashed black line in Fig.8 panels a-c."

Line 492: By "inter-model extend", are the authors referring to the range of simulated radar reflectivity values? Please clarify.

**The sentence was modified:**

"This range of simulated radar reflectivity values mainly reflects differences in the terminal velocities for different models and is comparable to the uncertainty of the ``MOSS'' hypothesis..."

Technical Corrections:

Line 16: Remove comma after "that".

**done**

Line 36: I don't believe "mid-latitudes" or "tropics" needs to be capitalized.

**corrected**

Lines 37, 40-41, and elsewhere: Remove parentheses around reference year.

**done**

Lines 51, 176, Table 1, and elsewhere: Change "ms-1" to "m s-1".

**changed**

Line 77: Add "the" before "DSD" and "PSD".

**added**

Line 97: Remove "a" before "X-".

**removed**

Line 106: Please add "the" before "methodology".

**added**

Line 255 and elsewhere: Change "kgm-2" to "kg m-2".

**Changed**

Line 336: Change "undergone" to "underwent"

**changed**

Line 448, 509: Change "approx." to "approximately"

**replaced**

Line 460: Change "even tighter relation. . . are expected" to "an even tighter relation. . . is expected"

**changed**

Line 462: Change "constrain" to "constraint"

**changed**

**Anonymous Referee #2**

Received and published: 30 August 2020

Overview:

The authors analyze data from ground-based triple-frequency radar, lidar, surface disdrometers and gauges, as well as environmental observations, with the aim of describing the vertical microphysical structure of liquid and ice-phase precipitation in a single, 6-hour period during which stratiform rain was observed. The data are of high quality and the methods involve optimal estimation of the precipitation PSD's using Doppler spectra and a collection of diverse particle models. The primary question posed is to what extent the flux of stratiform precipitation through the melting layer can be considered a steady, particle-mass-conserving process, and what microphysical mechanisms might lead to deviations from that kind of process?

The work is an original contribution, including the collected datasets which are fairly unique. The data and methods are generally described quite well, with good clarity of language, and the figures appropriately demonstrate the points made in the manuscript. However, there are some questions on the interpretation of the data and especially the "closure" procedure that will require some substantial explanation and/or revision, as detailed in Major Points, below.

The authors would like to thank for all the comments and the suggested corrections. In particular, the comment on the constraining the snow retrieval with the properties of rain made us realize that the validation of the MOSS assumption is not truly independent and, in fact, it is biased toward the mass flux continuity constraint. A new version of the algorithm, were any dependence between ice and rain is removed, is presented in the paper. Figure 9 shows the uncertainty of the retrievals that helps in the interpretation of the results. More details on the modifications introduced in the paper are listed below as the answers to the specific points.

**Major Points:**

(1) Section 3.4: Up through section 3.3, the manuscript is of high technical quality, and the authors' approaches and interpretations appear mostly sound. However, section 3.4 describes the optimal estimation (OE) of ice-phase particle properties that utilizes the Doppler spectrum to estimate ice-phase particle PSD's for different assumed ice particle models, selecting the most appropriate particle model based upon which one minimizes the OE's cost function.

In and of itself, the OE is fine. The problem is that the OE is constrained by (a) initial guesses, or priors, of the ice particle PSD's supplied by the Doppler-spectrum-derived rain PSD's which are extended to ice using the "melting only steady state" (MOSS) assumption, as well as (b) a second objective function term that constrains the ice and rain mass fluxes to be

closer (a difference fraction standard deviation of 0.33 is assumed). The MOSS assumption, in particular, is used to obtain a prior ice PSD that has the same mass flux as the rain below it. Clearly, the two prior terms (a) and (b), but especially (a), of the OE's objective function will tend to force the estimated mass fluxes of ice and rain to be more similar, regardless of the ice particle model chosen. But the primary purpose of the OE described in section 3.4 is to "assess the validity of the flux continuity assumption" as stated in the last sentence of that section. (The fact that the rain-spectrum-derived constraint is assigned a factor of two error doesn't really allow that much freedom to the OE solution, because as seen in Fig. 2b, e.g., a change of  $1 \times 10^{-4} \text{ m}^{-4}$  to  $2 \times 10^{-4} \text{ m}^{-4}$  in number density is not that large.)

Clearly, the application of such an OE could result in greater consistency of estimated ice and rain mass fluxes, and so as formulated, the estimated ice-phase precipitation fluxes from this OE can't be used to independently evaluate how much consistency there is between ice and rain fluxes. But that is precisely what is done in section 4.4. Unless I'm missing something, this is circular reasoning and not a scientifically valid approach.

If the authors want to address the ice vs. rain flux continuity issue in a quantitative way, they would need to decouple their rain and ice estimation procedures: What if no priors (referenced in a and b, above) are included in the objective function described in section 3.4, or what if only some simple gamma-fit to the ice particle Doppler spectrum is used as a prior? Either would decouple the rain and ice-phase estimation. If some prior based on rain-related PSD's and the MOSS assumption is required to get a stable estimate of ice PSD's, then one must question the information content of the ice Doppler spectrum and whether there is any way of independently estimating the ice PSD's and mass fluxes directly from their Doppler spectra.

The link between PSDs of ice and DSDs rain imposed in the retrieval was intended to stabilize the retrieval, but as you rightly pointed out, by using it the mass flux continuity conjecture cannot be evaluated independently. Therefore, we developed another version of the ice PSD retrieval where this dependence is removed. The properties of ice are derived independently for each considered snow model and the final estimate of the characteristic size of snow and the precipitation rate are derived as a weighted mean of the solutions corresponding to different models. The weight of each solution is computed as a softmax function of the distance between the measured and the simulated Doppler spectra. This is explained in Sect. 3.4. Section 4.3.2 has been modified to accommodate the changes due to the retrieval modification. Although, some numbers have changed the overall message remains the same because of similarities between the two snow retrieval results shown in Figure 1 below.

Figure 1. The full Doppler spectra retrievals applied both below and above the melting zone. Left: no dependence between the properties of ice and rain is used; right: PSD of ice constrained by the MOSS assumption

(2) p. 18, last paragraph of section 4.3.1, and p. 21 second paragraph: one of the difficulties of interpreting profile-type measurements is that one doesn't get a full 3D picture of the atmosphere, but just a 2D "curtain". Therefore, isn't it just possible that there was some horizontal variability of precipitation during the "aggregation" period and wind components perpendicular to the mean storm motion that could move aggregates of different concentrations into or out of the "curtain", so-to-speak? (At least evidence of vertical wind shear \*within\* the "curtain" is suggested by the tilted structures of Z and DFR in Figs. 1a and 1b, respectively.) The melting layer during the "aggregation" period had a depth of ~400 m, and so if the particles fell with an average speed of ~2 m/s, then they could potentially move laterally out of the 17 m wide radar beam in the ~200 s it took them to fall through the melting layer. If the precipitation was not strictly horizontally homogeneous, then that could cause difficulties for the authors' microphysical interpretation.

My general point here is that particle breakup in the melting layer is not the only possible explanation for higher ice-phase reflectivity fluxes relative to rain reflectivity fluxes during the "aggregation" period... all it would take is some horizontal variation of aggregates perpendicular to the "curtain" and some vertical variation of the horizontal wind.

Also, although breakup is certainly possible in the melting layer, melting aggregates could self-collect pretty efficiently as well.

The authors fully agree with this comment. This aspect was omitted in the discussion because the presented analysis is based on the 1D assumption due to unavailability of the data about the spatial variability of the system. However, we acknowledge that raising this point is important in the interpretation of the data we present. Therefore, the following discussion was added at the end of section 4.3.1:

One of the difficulties of interpreting profile-type measurements is that they do not provide a full 3D picture of the atmosphere, but just a 2D slice. Therefore, the presented conclusions are based on the assumption that the observed system is locally homogeneous i.e. despite horizontal winds the measurements taken below the melting layer correspond to the evolution of the ice PSD measured aloft. Considering the horizontal wind speed within the bright band (approx. 1.8 m/s during the "aggregation" period according to the ECMWF model) and the time needed for the particles to melt (approx. 3 minutes based on the MDV data) the precipitating system must be uniform over 325 m to meet this criterion. Because, the beam with of the X-band radar at the altitude of the melting zone is only 15 m, it is possible that the higher ice-phase reflectivity flux relative to rain can be a result of a horizontal gradient of the reflectivity that, for example, corresponds to the storm intensification along the wind direction. Note that, for the most part of the aggregation period the precipitation rate increases over time (see Fig. 9a) which supports this alternative interpretation.

**Minor Points:**

(3) Fig. 2 is a very informative reference, but some of the inset plots are very small and hard to read, particularly the snow spectrum panel above (C). Although these plots are meant to be symbolic, it would be good if they could be read more easily.

**Fonts of the labels were increased in the schematic that makes it much clearer.**

(4) p. 11, Eqs. (3) and (4), if v is meant to symbolize terminal velocity, shouldn't the capital V be used, as in Eqs. (1) and (2)? Also, I think w was previously defined in Eq. (1) as "negative upward". Shouldn't the w in Eq. (4) be similarly defined?

The velocity symbol, v, in the equations (3) and (4) was capitalized. The vertical wind is consistently defined as "negative upwards". The formula (4) just indicates that whenever non-zero wind is present the measured spectrum is shifted in the velocity domain. To avoid any confusion, an additional notation is introduced, i.e.  $S_{\lambda,w,target}$  denotes the reflectivity spectrum affected by the vertical winds.

(5) p. 15, beginning of first paragraph: when comparing the "aggregate" ice spectra to the "rimed" ice spectra in Fig. 5 (a) and (b) it looks like both the "aggregate" and "rimed" have mean peaks that are pretty steady in velocity up to 1.75 km altitude. The "aggregates" have a deeper structure that is more consistent, while the "rimed" particles peter out above 1.75 km and the peak becomes variable. It's a very minor point, but I would say 
[revised manuscript text omitted]
 [dBZdBZ], 2s-2s integration                                 | -50    | -70     | -58    |
| Nyquist Velocity $[\pm \frac{\text{ms}^{-1}}{\text{ms}^{-1}} \text{m s}^{-1}]$ | 80     | 10.5    | 10.2   |
| Range Resolution [mm]                                                          | 30     | 36      | 36     |
| Temporal Sampling [ss]                                                         | 2      | 2       | 3      |
| Lowest clutter-free range [mm]                                                 | 300    | 400     | 300    |
| Radome                                                                         | No     | No      | Yes    |

Depolarization Ratio (LDR) from the Ka-band radar following the method described in Devisetty et al. (2019). This approach is based on a very strong bright band signature in the LDR data in correspondence to the melting regardless of the rainfall intensity. In this study, the inflection points around the LDR peak are used as the top and the bottom of the melting zone. Over the presented time period, the altitude of the zero degree isotherm was very stable and decreased by only 300 m from

- 125 1.1 km at 6:00 to 0.8 km at 12:00. Radar reflectivity data below the bright band indicate two intervals of intensified rainfall: the first period is from 6:45 to 7:45 with the peak at 7:30, and a shorter interval that occurs around 9:00. Although for both periods similar X-band reflectivities are measured close to the ground (approx. approximately 27 dBZ), the reflectivity and the Dual Frequency Ratio (DFR) data suggest completely different ice microphysics aloft. The first period is characterised by larger X-band echos in the ice part coinciding with extremely large  $DFR^{X-Ka}$  values reaching 16 dB, which is a signature of
- 130 strong aggregation and presence of very large snowflakes (Kneifel et al., 2015). Almost no DFR is measured after 7:45 which indicates relatively small ice particles. Note that the DFR data in ice were corrected for attenuation prior to the analysis. The attenuation due to the rain was derived from the Rayleigh part of the dual-frequency spectral ratio (see e.g. Tridon et al., 2013) assuming negligible attenuation at the X-band. The extinction due to melting particles was estimated from the rainfall rates retrieved below the melting layer with the methodology of Matrosov (2008). This technique has been shown to be in agreement
- 135 with multi-frequency Doppler spectra estimates (Li and Moisseev, 2019). These two components were added together and were used as a path integrated attenuation correction factor that is applied to the column. This methodology does not account for any

---

## Author Response (AR2)

I would like to thank the reviewers for all the comments and the suggestions that greatly improved the quality of the paper. All the technical corrections were implemented.

Line 144: Change "back" to "black"

Line 215: Minor point, but can this be changed to V-D (instead of lowercase) for consistency with (2)? The same goes for line 251. Throughout the paper, I'd just like to see a consistent capitalization for terminal velocity between the equations and text regardless of whether v or V is ultimately used.

Line 242: Change "is" to "are"

Line 255: Change "morfe" to "more"

Line 450: Change "is" to "are"

Line 465: Change "an opposite" to "the opposite"